# Determining mean and standard deviation of the strong gravity prior through simulations

**Björn Jörges**[1]*, **Joan López-Moliner**[2]

**1** Center for Vision Research, York University, Toronto, ON, Canada, **2** Vision and Control of Action (VISCA) group, Department of Cognition, Development and Psychology of Education, Institut de Neurociències, Universitat de Barcelona, Barcelona, Catalonia, Spain

* bjoern_joerges@hotmail.de

**Data Availability Statement:** The data are available on GitHub (github.com/b-jorges/SD-of-Gravity-Prior).

**Funding:** Funding was provided by the Catalan government (2017SGR-48; https://govern.cat/gov/

## Abstract

Humans expect downwards moving objects to accelerate and upwards moving objects to decelerate. These results have been interpreted as humans maintaining an internal model of gravity. We have previously suggested an interpretation of these results within a Bayesian framework of perception: earth gravity could be represented as a Strong Prior that overrules noisy sensory information (Likelihood) and therefore attracts the final percept (Posterior) very strongly. Based on this framework, we use published data from a timing task involving gravitational motion to determine the mean and the standard deviation of the Strong Earth Gravity Prior. To get its mean, we refine a model of mean timing errors we proposed in a previous paper (Jörges & López-Moliner, 2019), while expanding the range of conditions under which it yields adequate predictions of performance. This underscores our previous conclusion that the gravity prior is likely to be very close to 9.81 m/s². To obtain the standard deviation, we identify different sources of sensory and motor variability reflected in timing errors. We then model timing responses based on quantitative assumptions about these sensory and motor errors for a range of standard deviations of the earth gravity prior, and find that a standard deviation of around 2 m/s² makes for the best fit. This value is likely to represent an upper bound, as there are strong theoretical reasons along with supporting empirical evidence for the standard deviation of the earth gravity being lower than this value.

## Introduction

There is ample evidence that humans represent earth gravity and use it for a variety of tasks such as interception [1–10], time estimation [11], the perception of biological motion [12] and many more. Recently, we have shown that gravity-based prediction for motion during an occlusion matched performance under a 1g expectation not only qualitatively, but also quantitatively [13]. This was an important finding to support our interpretation of the above results as a strong prior in a Bayesian framework of perception [14]. The results presented in [13] indicate that temporal errors in a timing task were consistent with a mean of 1g (9.81 m/s²) when occlusions were long enough. In the present paper, we extend the simulations brought forward in our previous paper: First, we consider how accounting for the Aubert-Fleischl effect, which leads humans to perceive moving object at about 80% of their actual speed when

) and the European Regional Development Fund's (https://ec.europa.eu/regional_policy/en/funding/erdf/) project ref. PSI2017-83493-R from AEI/Feder, UE. BJ was supported by the Canadian Space Agency (CSA). The funders had no role in study design, data collection and analysis, decision to publish, or preparation of the manuscript.

**Competing interests:** The authors have declared that no competing interests exist.

they pursue the target with their eyes [15–17], can extend our simple 1g-based model to shorter occlusions. Furthermore, to fully characterize a prior, we need to not only indicate its mean, but also its standard deviation. The second goal of the present paper is thus to determine the standard deviation of the strong gravity prior. We aim to achieve this goal by simulations based on assumptions about the different sources of noise relevant to the task at hand.

In this paper, we adopt a constructivist-computational framework [18, 19]; we view perception as a process by which humans acknowledge the state of the world around us based on both prior knowledge and sensory online information in order to guide their interactions with the external world. Please note that other psychological traditions, such as ecological perception [20], deny the necessity of prior knowledge. Within our constructivist framework, we envision (visual) perception as a two-step process: Encoding and Decoding [21, 22]. During Encoding, low level signals such as luminosity, retinal velocities or orientation are picked up by the perceptual system and represented as neural activity. However, these low-level sensory signals, and the neural activity they are represented as, can be ambiguous with respect to the state of the world: for example, the same retinal velocities can correspond to vastly different physical velocities, depending on the distance between observer and object. An object that moves 6 m in front of the observer in the fronto-parallel plane with a physical speed of 1 m/s elicits a retinal speed of about 9.5˚/s when fixation is maintained. The same retinal speed could correspond to a target that moves at a physical speed of 1.2 m/s 7 m in front of the observer. Decoding is the process of interpreting optic flow information. In Decoding, humans often combine sensory input with previous (prior) knowledge to obtain a more accurate and precise estimate of the observed state of the world. For example, we use knowledge about the size of an object to recover its most likely distance to the observer, thus providing a key to recover its physical velocity from retinal motion. If we, for example, know that we are observing a basketball and know from experience that its radius is 0.12 m, and we perceive that the target occupies a visual angle of 0.5˚, we know that the target moves at 7 m in front of as. We then also know that the physical velocity of the ball is 1.2 m/s, not 1 m/s. In some, if not many instances, this combination occurs according to Bayes' formula:

$$P(A|B) = \frac{P(B|A)P(A)}{P(B)} \qquad [1]$$

The probability of a state of the world A given evidence B is the probability of observing evidence B given the state of the world A multiplied by the probability of the state of the world (A), divided by the probability of the evidence (B). In a Bayesian framework, sensory input (Likelihood), corresponding to the term $\frac{P(B|A)}{P(B)}$ in Eq 1, and prior knowledge (Prior), corresponding to $P(A)$ in Eq 1, are combined according to their respective precisions to yield a more precise and more accurate final percept (Posterior). Under many circumstances, Prior, Likelihood and Posterior can be represented as normal distributions whose standard deviations correspond to the representation's reliability. If an organism has a high sensitivity to the sensory input, that is, when they are able to reliably distinguish one stimulus strength from a very similar stimulus strength, the standard deviation of the Likelihood would be very low, which corresponds to a very narrow distribution. On the other hand, if the organism has a very precise representation of the most likely state of the world, the Prior would be very narrow. Finally, the standard deviation of the Posterior would depend on the precision of Likelihood and Prior. Usually, both the Prior and the Likelihood contribute to the Posterior; for example when we know that our opponent in a tennis match *usually* serves in the right corner of the court, but *not always*, (Prior) and we have good visibility of their serving motion, but since the motion is so quick, we do not have a lot of time to acquire evidence (Likelihood). We

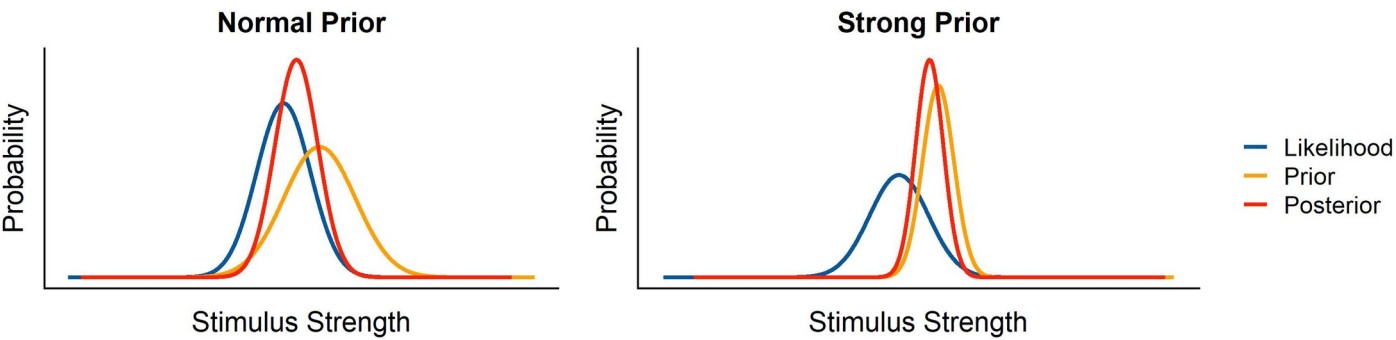

**Fig 1. Graphical illustration of likelihood, prior and posterior in a Bayesian framework, for both a normal, relatively shallow prior, and a strong, extremely precise prior.**

thus take sensory input (e. g. about their body posture while serving) into account only to some extent (see "Normal Prior" scenario in Fig 1). However, in the case of gravity it seems that the expectation of Earth Gravity overrules all sensory information that humans collect on the law of motion of an observed object [6, 7, 23–25]. On a theoretical level, this is a sensible assumption, since all of human evolution and each human's individual development occurred under Earth Gravity. In Bayesian terms, the Prior is extremely precise and thus overrules all sensory information represented as the Likelihood. According to our interpretation, we would thus expect an extremely low value for the standard deviation of the earth gravity prior ("Strong Prior" scenario in Fig 1). We would expect this value to be represented more precisely than linear velocities, which generally elicit Weber Fractions of 10%, which corresponds to a standard deviation of about 15% of the mean represented stimulation.

In the following, we use the data from our previous study [13] to simulate the variability of responses under different assumptions about the standard deviation of the gravity prior.

## Methods

In this paper, we use previously published data [13]. The pre-registration for the original hypotheses can view viewed on Open Science Foundation (https://osf.io/8vg95/). All data relevant to this project are available in our GitHub repository (https://github.com/b-jorges/SD-of-Gravity-Prior).

### Participants

We tested ten participants (n = 10) overall, including one of the authors (BJ) who was excluded from the analyses in this paper. The remaining participants were between 23 and 34 years old and had normal or corrected-to-normal vision. Three (n = 3) of the included participants were women and six (n = 6) were men. All participants gave their informed consent. The research in this study was part of an ongoing research program that has been approved by the local ethics committee of the University of Barcelona. The experiment was conducted in accordance with the Code of Ethics of the World Medical Association (Declaration of Helsinki).

### Stimuli

Participants were shown targets of tennis ball size (r = 0.033), shape and texture in an immersive 3D environment (see Fig 2). The 3D environment should help participants to perceive the stimulus at the correct distance and activate the internal model of gravity [11]. The targets moved along parabolic trajectories in the fronto-parallel plane 6.15 m in front of the observer.

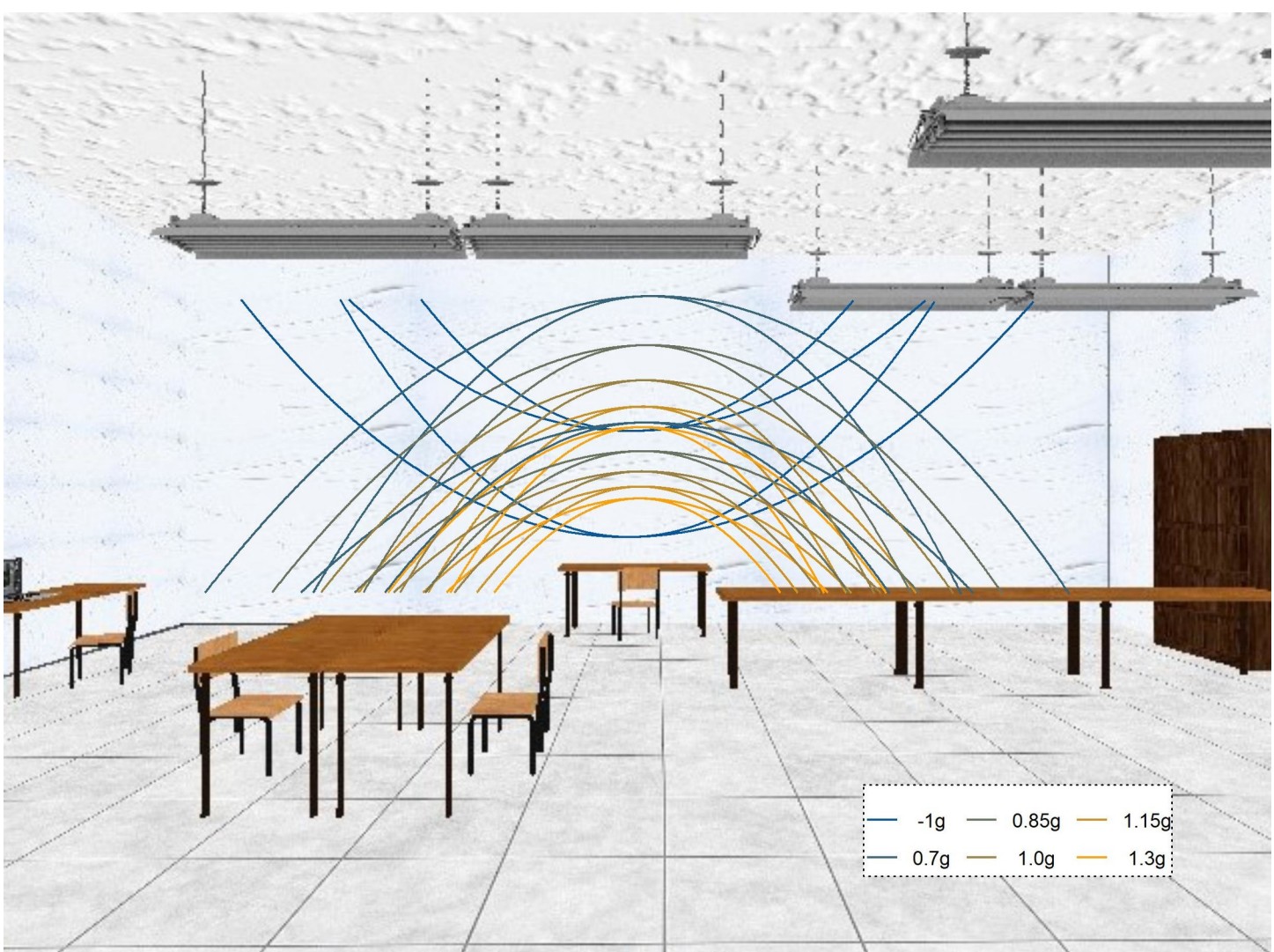

**Fig 2. 2D depiction of the visual scene used as environment for stimulus presentation.** The stimulus was always presented in front of the white wall and never crossed other areas (such as the lamps of tables) that could introduce low level differences in contrast etc. The lines denote the different parabolic trajectories that along which the targets travelled. Figure from (Jörges & López-Moliner 2019).

The trajectories were determined by the simulated gravity (0.7g, 0.85g, 1g, 1.15g, 1.3g or -1g), the initial vertical velocity (4.5 or 6 m/s) and the initial vertical velocity (3 or 4 m/s). Air drag was simulated according to Eqs [2] and [3] (see http://www.demonstrations.wolfram.com/ProjectileWithAirDrag/) in line with the air drag at the location of the experiment (Barcelona in Spain, at sea-level), and the ball did not spin.

$$x(t) = \frac{(v_{xi}^2 + v_{yi}^2)^{0.5} * m * g}{g * c} * \cos\left(\text{asin}\left(\frac{v_{yi}}{(v_{xi}^2 + v_{yi}^2)^{0.5}}\right)\right) * \left(1 - e^{\left(-\frac{g*t*c}{m*g}\right)}\right) \quad [2]$$

$$y(t) = \left(\frac{m}{c}\right) * \left((v_{xi}^2 + v_{yi}^2)^{0.5} * \sin\left(\text{asin}\left(\frac{v_{yi}}{(v_{xi}^2 + v_{yi}^2)^{0.5}}\right)\right) + \frac{m * g}{c}\right) * \left(1 - e^{\left(-\frac{g*t*c}{m*g}\right)}\right)$$
$$- \frac{m * g * t}{c} \quad [3]$$

$x(t)$ is the horizontal position over time, $y(t)$ is the vertical position over time, $v_{xi}$ is the initial horizontal velocity, $v_{yi}$ is the initial vertical velocity, $m$ is the mass of the object (0.057 kg), $g$ is the simulated gravity, $c$ is the drag coefficient (0.005). Targets always moved from left to right. When gravity acted downwards, the target started 0.5m above the simulated ground and when it acted upwards, the target started out 3.5m above the ground. The final positions were marked with tables for downwards gravities and by lamps hanging from the ceiling for upwards gravities. The total flight time was the time it took for the ball to return to its initial height. The target disappeared either between 75% and 80% (Short Occlusion) or between 50% and 55% (Long Occlusion) of the total flight time. Each of the conditions was repeated 24 times, for a total of 1344 trials across four blocks. Within each block, the kinetic profiles were presented in a random order. From the participant's perspective, the trajectories always unfolded in front of the white wall, that is, low level cues such as contrast and brightness were equal across all trajectories and conditions. Fig 2 shows the trajectories projected on the visual scene.

## Apparatus

We used two Sony laser projectors (VPL-FHZ57) to present overlaid images on a back-projection screen (244 cm high and 184 cm wide). The images had a resolution of 1920 x 1080 pixels and were refreshed at 85Hz. Participants were wearing glasses with polarizing filters to provide stereoscopic images. They stood 2 m in front of the screen. The disparity between the two projectors' images was adapted to each participant's interocular distance. The stimuli were programmed in PsychoPy [26]. The projectors introduced a delay of 0.049259 s (SD = 0.001894 s) that we accounted for in the analysis of timing responses. For another hypothesis, eye-tracking data was acquired; see [13].

Participant responses were collected with a regular computer mouse. It has been shown that commodity input devices often lack in temporal accuracy and precision for response capture [27]. To mitigate such issues, we use the openGl engine in python (pyglet) devoted to gaming, which aims to reach maximum precision both for stimulus frames and input recording. We access the mouse time stamps directly iohub python libraries (which merges with PsychoPy) which circumvents the main system events loop and uses the clock_gettime (CLOCK_MONOTONIC) in unix-like systems (like os x, the one we use). The precision is sub-milliseconds. Iohub can be used with or without PsychoPy real-time access to input devices. Importantly, it runs its own thread devoted to continuously sampling the input device state independently of the video (stimulus) thread.

## Procedure

We asked participants to follow the target closely with their gaze and indicate with a mouse click when they believed the target had returned to its initial height. Participants first completed 48 familiarization trials in which the balls reappeared when they pressed the button, which allowed them to assess the spatial error. Then, the main experiment followed. It consisted of four blocks: 3 blocks with 320 trials each (the five positive gravities– 0.7g, 0.85g, 1g, 1.15g, 1.3g –, two initial vertical velocities, two initial horizontal velocities, two occlusion conditions, eight repetitions per condition) and one block with 384 trials (as the other block, but 1g and -1g as gravities, and 24 repetitions per condition). Each block took 15–20 minutes and participants could rest after each block. We counterbalanced across participants whether the -1g block or the 0.7g-1.3g blocks was presented first.

## Results

We have reported mean difference in a previous paper [13]. In the following, we thus limit ourselves to analyzing the influence of gravity on the *precision* of responses in preparation for the simulations we are conducting after. We used a slightly different, more liberal outlier analysis for this project to make sure that we do not lose any variability present in participants' responses. We also exclude all data collected from the author (s10; all 1344 trials). Further, we exclude all trials where subjects pressed the button before the target disappeared (38 trials) or where the temporal error was greater than 2 s (178 trials). Overall, we excluded 1.6% of all trials from the nine participants included in the analysis. To make it easier to compare temporal errors across conditions, we then computed the error ratio:

$$Error\ Ratio = \frac{Error + Occluded\ Duration}{Occluded\ Duration} \tag{4}$$

In Fig 3, we illustrate the response distributions. For an analysis and interpretation of the effect of gravitational motion on accuracy, please see our previous paper [13].

While we used Linear Mixed Modelling to assess accuracy, assessing precision differences between conditions is not straight-forward with this method. Therefore, we employ Bayesian Linear Mixed Modelling to assess whether gravity has an impact on the precision of the timing responses. The R package brms [28], which provides user-friendly interface for the package rstan [29], uses a very similar syntax to the more well-known lme4 [30]. In addition to mean differences, this type of analysis also allows us to test for variability differences between conditions. We thus fit a mixed model to explain both means and standard deviations of the response distributions, with gravity as a fixed effect and varying intercepts per participant as random effects. In lme4/brms syntax, the test model is specified as:

$$Error\ Ratio \sim Gravity + (1|Subject)$$
$$Sigma \sim Gravity + (1|Subject) \tag{5}$$

Where the first line corresponds to the statistical structure that corresponds to the means of the response distributions and the second line corresponds to the standard deviations of the response distributions. Unlike regular Linear Mixed Models, Bayesian Linear Mixed Models do not need to be compared to a Null Model. We can use the hypothesis() function from the R Core package [31] to test hypotheses directly. We found a posterior probability of >0.999 that a lower gravity value is related to lower variability, the sigma coefficient for Gravity being 0.057 (SE = 0.004; 95% Confidence Interval = [0.051;0.064]) in the log space. In the regular space, this corresponds to a standard deviation of 0.296 (95% CI = [0.282;0.313]) for 0.7g, 0.321 (95% CI = [0.303;0.344]) for 0.85g, 0.350 (95% CI = [0.326;0.378]) for 1g, 0.382 (95% CI = [0.351;0.416]) for 1.15g and 0.413 (95% CI = [0.378;0.458]) for 1.3g. Table 1 lists all mean temporal errors and the respective standard errors across participants. Note that, unlike the results from the Bayesian Mixed Model, the variability values from Table 1 also include variability that the Mixed Model assigns to the individual.

Interestingly, precision seems to be higher for 1g trials than for -1g trials. To test this observation statistically, we fitted a second Bayesian Linear Mixed Model to the -1g/1g data, where gravity as fixed effect factor and subjects as random effects predict the timing error:

$$Error\ Ratio \sim Gravity + (1|Subject)$$

We tested the hypothesis that Gravity would lead to lower variability. The posterior probability of this hypothesis being true was > 0.999, with a sigma coefficient for Gravity of -0.011 (SE = 0.004; 95% Confidence Interval = [-0.014,-0.009] in the log space. That is, the standard

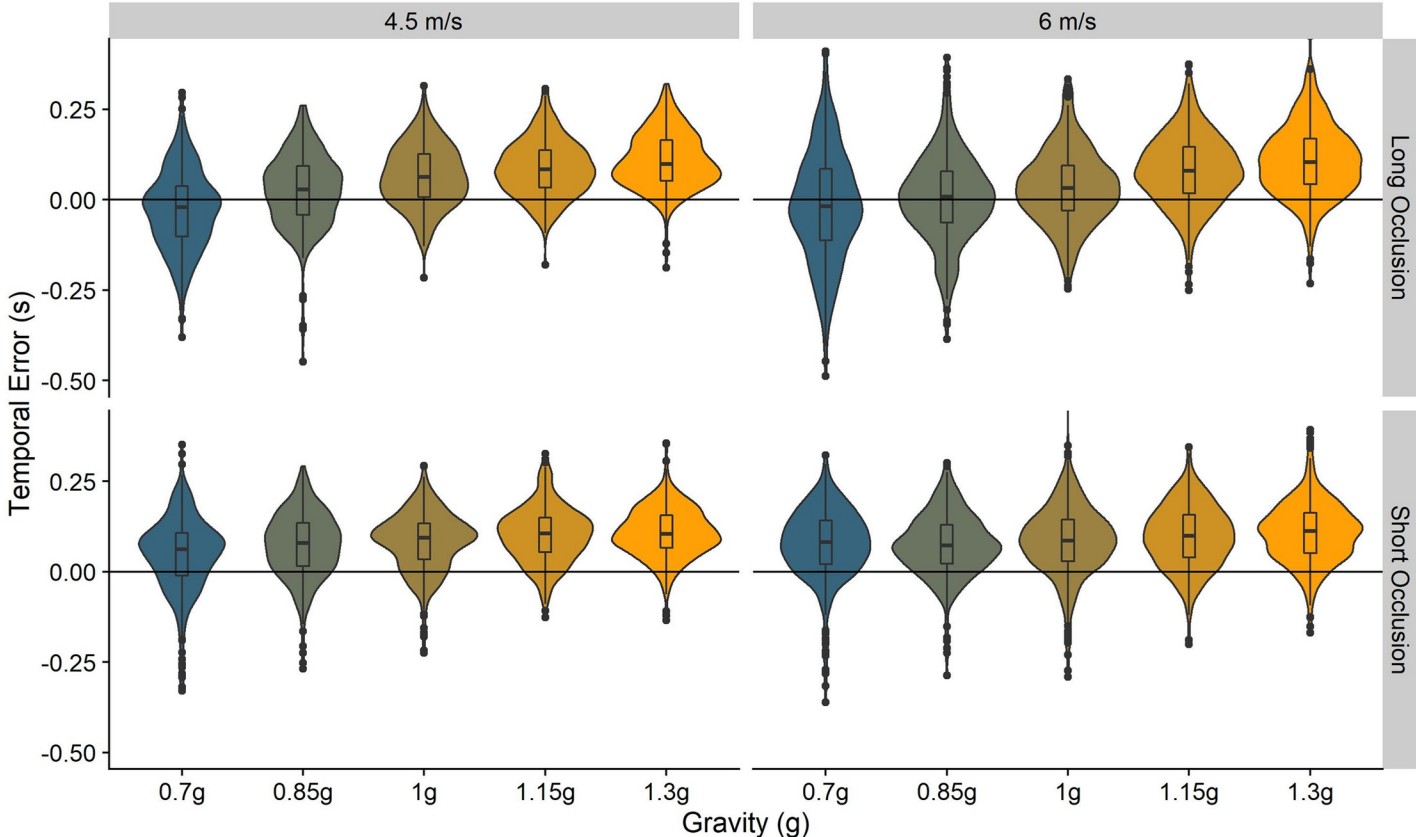

**Fig 3. Temporal errors in the 0.7–1.3 g conditions.** The wings of each structure indicate the distribution of responses, while the boxplot in the middle of each structure indicate the 75% percentiles and the mean per condition.

deviation of distribution of -1g responses in regular space is 0.426 (95% Confidence Interval = [0.414;0.439]), while the standard deviation of the distribution of 1g responses in regular space is 0.344 (95% Confidence Interval = [0.334;0.353]). This indicates that the absolute error is lower and thus the precision is higher for 1g than for -1g. On a theoretical level, this is in line with previous findings [32] showing that the internal representation of gravity is not activated

**Table 1. Means and standard deviations observed for the temporal errors divided by gravities and initial vertical velocities.**

| | | | 0.7g-1.3 Block | | | | | -1g/1g Block | |
|---|---|---|---|---|---|---|---|---|---|
| | | | **Long Occlusion** | | | | | | |
| $v_{yi}$ | | | *0.7g* | *0.85g* | *1g* | *1.15g* | *1.3g* | *-1g* | *1g* |
| **4.5 m/s** | *Mean* | | 1.12 | 1.11 | 1.20 | 1.24 | 1.30 | 1.33 | 1.17 |
| | *SD* | | 0.47 | 0.49 | 0.53 | 0.42 | 0.44 | 0.53 | 0.38 |
| **6 m/s** | *Mean* | | 1.05 | 1.11 | 1.17 | 1.24 | 1.32 | 1.23 | 1.16 |
| | *SD* | | 0.49 | 0.55 | 0.57 | 0.54 | 0.57 | 0.56 | 0.46 |
| | | | **Short Occlusion** | | | | | | |
| $v_{yi}$ | | | *0.7g* | *0.85g* | *1g* | *1.15g* | *1.3g* | *-1g* | *1g* |
| **4.5 m/s** | *Mean* | | 1.22 | 1.31 | 1.34 | 1.41 | 1.52 | 1.68 | 1.35 |
| | *SD* | | 0.64 | 0.65 | 0.65 | 0.56 | 0.88 | 0.86 | 0.58 |
| **6 m/s** | *Mean* | | 1.26 | 1.33 | 1.37 | 1.47 | 1.49 | 1.51 | 1.35 |
| | *SD* | | 0.65 | 0.77 | 0.77 | 0.88 | 0.75 | 0.80 | 0.76 |

when upwards motion is presented, even when the absolute value of acceleration impacting the object is equal to the absolute value of earth gravity (9.81 m/$^2$). The precision may thus be higher for 1g than for -1g because the internal model of gravity is utilized for 1g, but not for -1g trials.

## Simulations

The physical formula for distance from initial velocity and acceleration (Eq 6) is the base for both of our simulation procedures. This reflects the assumption that humans perform the task at hand accurately–under most circumstances. This assumption is supported by our data, which show a high accuracy for the earth gravity conditions.

We furthermore neglect the air drag for these simulations and use the equation for linearly accelerated motion as an approximation.

$$d_y = \frac{g}{2} * t^2 + v_y * t \qquad [6]$$

$$t_{1/2} = \frac{-v_y {}^{+}_{-} \left( v_y{}^2 - 4 * \frac{g}{2} * d_y \right)^{0.5}}{2 * \frac{g}{2}} \qquad [7]$$

As evidenced by a comparison between Eqs (2) and (3) and Eqs (6) and (7), the computational complexity increases significantly if we want to accommodate air drag, while the gains in accuracy are marginal (0.02 s in the condition with the most extreme differences).

### Mean of the gravity prior

To characterize the mean Strong Gravity Prior, we build upon our model the mean timing errors presented in our previous data [13]. Importantly, the predictions of our model matched the observed data only for the Long Occlusion condition. In the Long Occlusion condition, subjects displayed a tendency to respond slightly too late, while their responses should be centered around zero. Our ad hoc explanation of this discrepancy was that subjects were often executing a saccade when the ball returned to initial height, which may have interfered with the predictions [33]. An alternative explanation may be, however, that our subjects underestimated the target's speed at disappearance due to the so called Aubert-Fleischl phenomenon: humans estimate the speed of a target that they pursue with their eyes at about 80% of its actual speed [15, 16, 34–36]. Our subjects were specifically instructed to follow the target with their eyes, and the eye-tracking data we collected that they generally did pursue the target [33]. An underestimation of the velocity at disappearance could explain the tendency of subjects to respond too late in the Short Occlusion condition. For the Long Occlusion condition, on the contrary, the vertical speed at disappearance is very low and has a nearly neglectable influence on the final prediction. Setting the perceived velocity at 80% of the presented velocity should thus yield more accurate predictions for the Short Occlusion condition, while the accuracy for the Long Occlusion condition would be largely maintained. We thus employ the same procedure laid out in [33], but add a coefficient of 0.8 to the perceived velocity at disappearance to account for the Aubert-Fleischl phenomenon.

We will briefly summarize the procedure and then present how this tweak affects the results of our simulations. We used the physical formula for distance from accelerated motion (Eq 6, with $d$ being the height as disappearance, $v_y$ the vertical velocity at disappearance and $g$ being gravity). For our simulations, we assume that humans use an earth gravity value of 9.81 m/s$^2$ independently of the presented gravity value, as long as the display is roughly in line with a

real-world scenario. We furthermore assume that we perceive the vertical velocity at disappearance at 80% of the presented velocity. Eq 7 thus becomes

$$t_{1/2} = \frac{-v_{y,perceived} \pm_{-}^{+} \left( v_{y,perceived}^2 - 4 * \frac{g_{earth}}{2} * d_y \right)^{0.5}}{2 * \frac{g_{earth}}{2}} \qquad [8]$$

With $v_{y,perceived} = 0.8 * v_{y,presented}$ and $g_{earth} = \frac{9.81 \, m}{s^2}$.

We use this formula to simulate the timing error for each trial separately without adding noise. We furthermore also simulate the responses without accounting for the Aubert-Fleischl phenomenon to compare performance for both models. Fig 4 shows the mean errors observed in our participants ("Obs. Error"), the mean errors when accounting for the Aubert-Fleischl

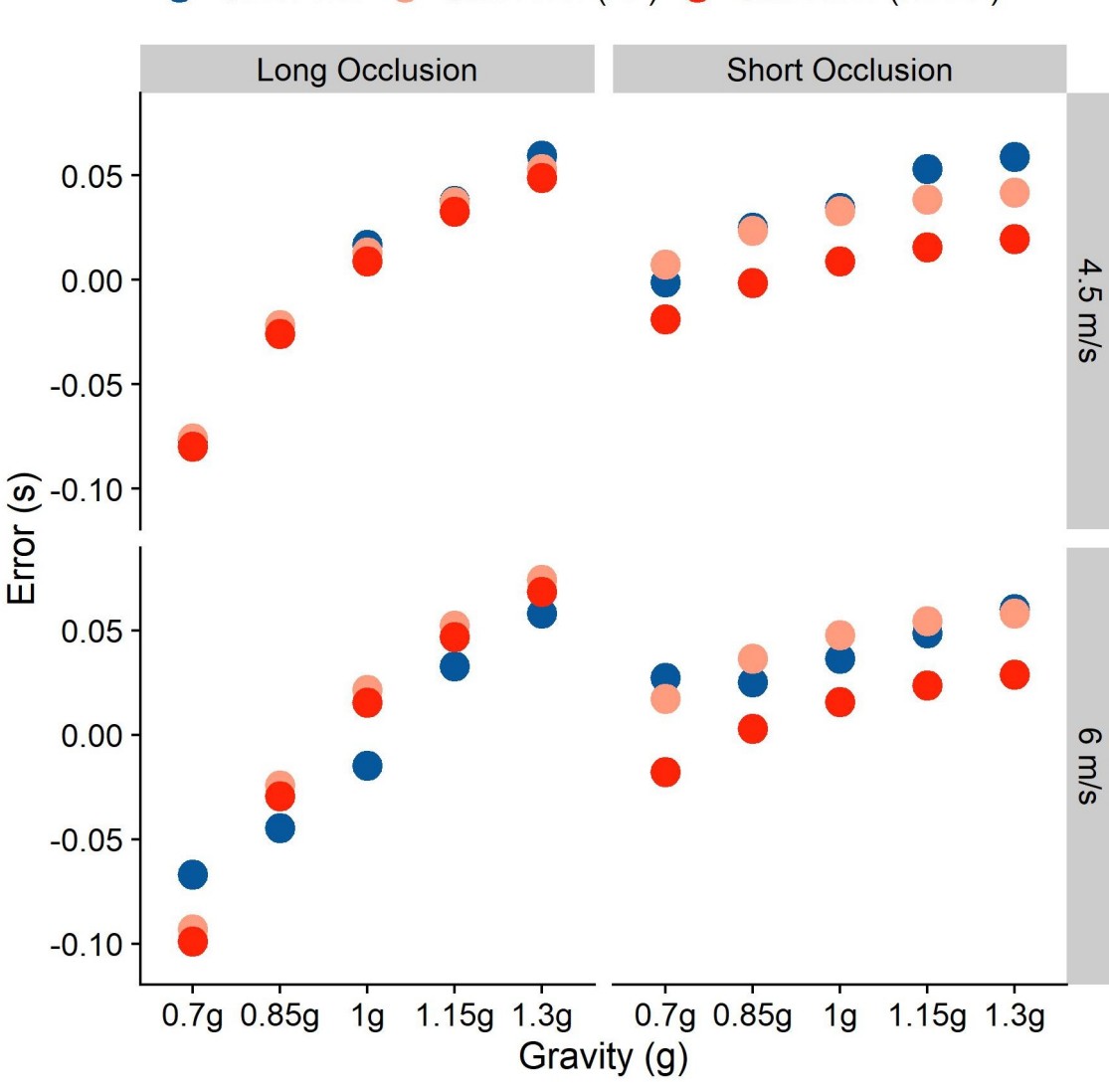

**Fig 4. Mean temporal errors that we observed in our participants (across participants in blue, and for each participant separately in shades of grey), simulated taking the Aubert-Fleischl phenomenon into account (light red) and simulated without taking the phenomenon into account for the different conditions.** The right column represents values for the Long Occlusion condition, while the left column represents the Short Occlusion condition. The upper row shows values for an initial vertical velocity of 4.5 m/s, while the lower row represents initial vertical velocities of 6 m/s. Note that the standard errors for the observed errors are so small that all error bars fall well within the area covered by the dots.

**Table 2. Root Mean Squared Errors (RMSEs) between simulated and observed mean errors for simulations including the Aubert-Fleischl phenomenon (AF) and simulations that don't (No AF).** Lower values signify a better fit.

| $v_{yi}$ | Long Occlusion | | Short Occlusion | |
|---|---|---|---|---|
| | AF | No AF | AF | No AF |
| 4.5 m/s | 0.150 | 0.160 | 0.236 | 0.333 |
| 6 m/s | 0.148 | 0.158 | 246 | 0.344 |

phenomenon ("Sim. Error (AF)"), and the mean errors when not accounting for the Aubert-Fleischl phenomenon ("Sim. Error (No AF)").

The overall Root Mean Squared Error between AF model predictions and observed behavior is 0.2, and for the non-AF model predictions substantially higher, at 0.265. Table 2 shows the error for each of the conditions. Including the AF phenomenon thus vastly improves the model's generalizability.

This improvement upon our previous model lends further support to the idea that the mean of a strong gravity prior is at or very close to 9.81 /s$^2$.

## Standard deviation of the gravity prior

The second value needed to characterize a normal distribution, which we assume the strong gravity prior to be represented as, is its standard deviation. There are two different ways to approach this problem: First, we can simulate the temporal responses of our subjects assuming different standard deviations for the gravity prior and minimize the difference between the standard deviations of the responses we observed in our subjects and the model standard deviations. In this case, we would draw the values for $v_y$, $d_y$ and $g_{earth}$ from distributions with given means and standard deviations, and compute a simulated temporal response from these values. The mean for $v_y$ would be the last observed velocity in y direction, corrected by a factor of 0.8 for the Aubert-Fleischl phenomenon, and the standard deviation can be computed based on Weber fractions for velocity discrimination from the literature. The mean for $d_y$ is the distance in y direction between the point of disappearance and the reference height. The mean for $g_{earth}$ is 9.81 m/s$^2$, and we optimize over its standard deviation to match the standard deviation observed in the subjects' temporal responses.

A second approach would be to solve Eq (6) for $g_{earth}$, and then compute its mean and standard deviation analytically based on the means and standard deviations of $t$, $v_y$ and $d_y$. For the addition, subtraction and multiplication of two normal distributions, there are analytic solutions to compute mean and standard deviation of the resulting distribution.

$$g_{earth} = \frac{2(d_y - v_y * t)}{t^2}$$

[9]

However, as evident from Eq 9, this method requires computing the standard deviation of the quotient of two distributions. To our knowledge, this is not possible in an analytical fashion and would entail simulations by itself. We will thus focus on the simulation approach.

**Assumptions.** For this approach, we need to make several assumptions. In the following, we will outline each and provide the rationale for the chosen values. Please note that we conduct these simulations in absolute terms (i.e., absolute errors) to mimic the processes more closely, but convert quality metrics (such as model fits) and results into relative terms (i.e., error ratios).

*Use of Eq (6).* In our previous paper, we have shown that predictions based on Eq 6 fit observed temporal errors reasonably well [13]. This is particularly the case when subjects

extrapolated motion for larger time frames in the Long Occlusion condition. The difference in predictions for this equation with regards to Eq (2) is at most 3 ms, and the added computational complexity does not justify the added accuracy, especially since our main concern is precision.

$v_y$. The velocity term in Eq 6 ($v_y$*t) refers to the part of the full distance the target moved because of its initial velocity. Our targets disappeared right after peak, therefore their initial velocity was very low. The velocity term thus contributes less to the full estimate than the gravity term, especially in the Long Occlusion condition (see also Fig 5C). Importantly, the vertical velocity component is not perceived directly. Rather, it has to be recovered from the tangential speed ($v_{tan,perceived}$) and the angle between the tangential speed vector and the vertical speed vector ($\alpha_{perceived}$) by means of the equation:

$$v_{y,perceived} = \cos\left(\alpha_{perceived}\right) * v_{tan,perceived} \tag{10}$$

Weber fractions for the discrimination of angular velocities reported in the literature are about 10% [37]. To calculate the standard deviation of the distribution of perceived velocities from the Weber fraction, we have to find that normal distribution where a difference of 10% from its mean leads to a proportion of responses of 25/75%. For a standardized normal distribution with a mean of 1, this is a standard deviation of 0.148. Note that, by using a standardized normal distribution, we assume that Weber fractions are constant across the relevant range of stimulus strengths. Fig 5C shows how predictions vary with varying variability in perceived vertical velocity: The effect is negligible for the Long Occlusion condition, while it increases response variability uniformly across gravities. Further variability is incurred in estimating $\alpha_{perceived}$. Following [38], the JND for orientation discrimination in untrained subjects is around 6° for oblique orientations. This corresponds to a standard deviation of 0.089.

Furthermore, we need to account for the Aubert-Fleischl phenomenon, which consists in an underestimation of the velocity of a moving target during smooth pursuit [15, 16, 34–36]. While this effect should in principle be partially offset by improved predictions for motion coherent with earth gravity – an empirical question that has, to our knowledge, not been addressed so far –, our simulations show that a Aubert-Fleischl correction factor of 0.8 yields an excellent fit for the observed mean errors. We thus proceed with a value of 0.8 also for the simulations concerning the standard deviation.

$d_y$. For the distance term ($d_y$), we choose the stimulus value as mean distance, as we don't expect any biases. In terms of precision, Weber fractions of 3% to 5% are observed for distance estimates in the front parallel plane [39]. However, since subjects have to estimate the distance not between two well defined points, but rather the height above the simulated table, the precision of these estimates is likely lower than reported for the above task. We thus work with a Weber fraction of twice the reported value (10%). Using the above method, we determine that the standard deviation for this value is 0.148. Fig 5A shows how predictions vary with variability in perceived distance: There is a slight logarithmic pattern, where response variability added by higher variability in perceived distance increases with decreasing gravity.

$t$. The response time t is measured directly in our task, both in mean and variability.

**Remaining variability.** For our simulations, we rely on accounting for every source of variability in the responses. One source of error beyond perceiving and representing **g**, $v_y$ and $d_y$ is the motor response. Motor responses are likely to vary strongly between tasks, for which reason variability reported in the literature is of limited use. To estimate the error introduced by these further factors, we thus take advantage of previous results indicating that the gravity

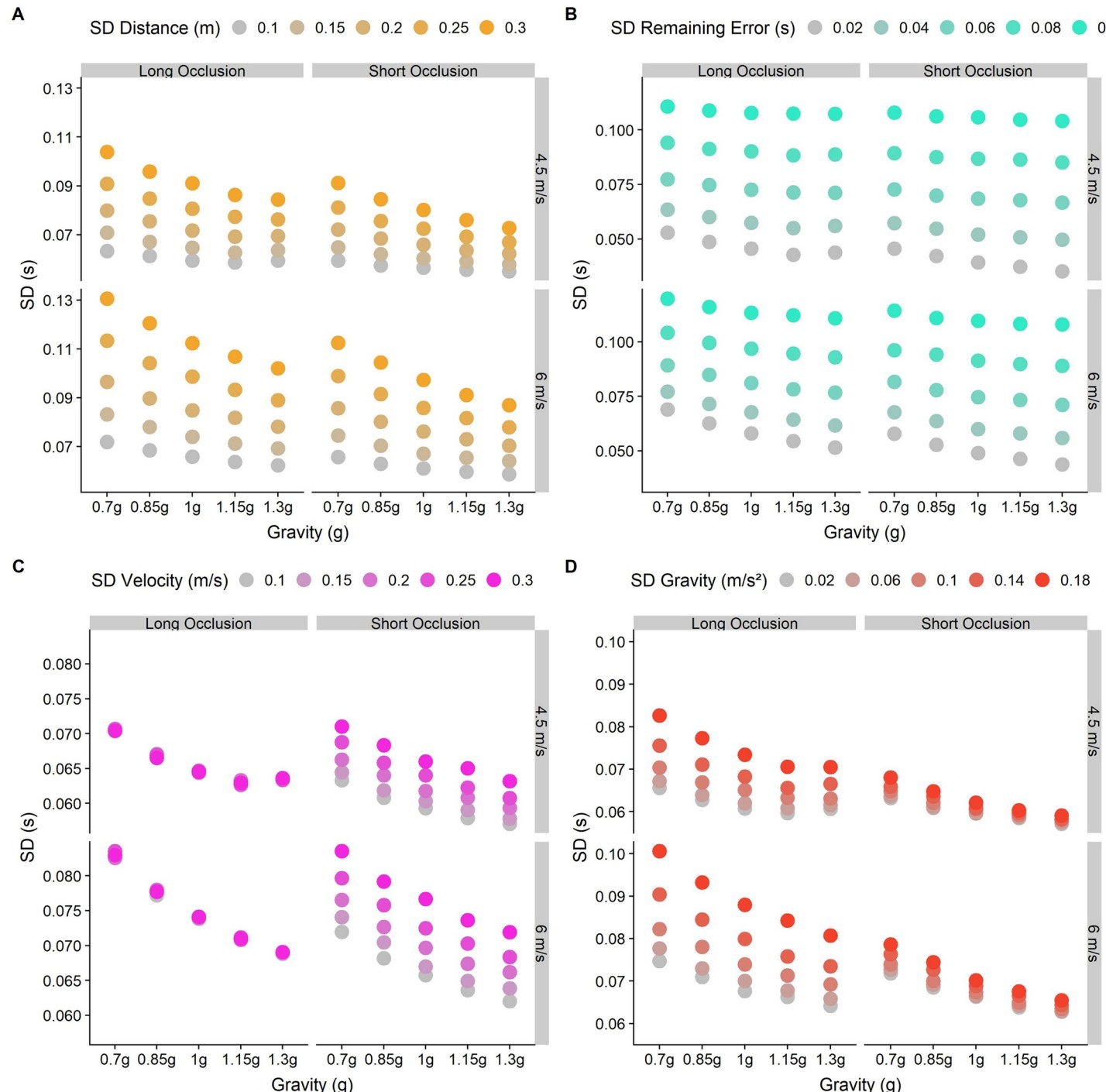

**Fig 5. Predictions for different standard deviations chosen for different parameters in our model.** Dots represent the standard deviation for each gravity (0.7g-1.3g), divided by Occlusion category (Long and Short) and initial vertical velocities (4.5 and 6 m/s). The color gradient indicates different values of the (standardized) standard deviation for the perceived distance, the perceived velocity, the represented gravity and the remaining error. The baseline values are 0.148 for distance and velocity, 0.1 for gravity and 0.05 for the remaining (motor) error. A. Predictions for five standardized standard deviations for the perceived distance (0.1–0.3 m). B. Predictions for five standard deviations for the remaining (motor) error (0.02–0.1 s), modelled as independent of and constant across initial velocities, gravities and occlusion conditions. C. Predictions for five different standardized standard deviations for the last perceived velocity (0.1–0.3 m/s). D. Predictions for five different standardized standard deviations for the represented gravity (0.02–0.18 m/s²).

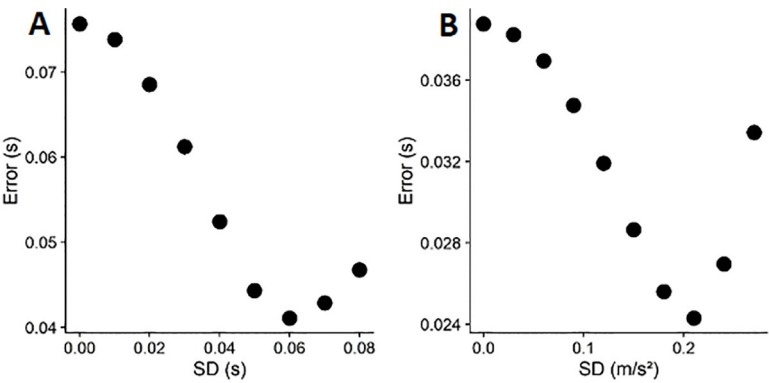

**Fig 6.** A. Root Mean Square Errors (RMSE) between the standard deviation of timing errors simulated based on different motor errors (between 0.00 and 0.07 s) and the standard deviation of observed timing errors. B. Root mean square errors (RMSE) between the standard deviation of timing errors simulated based on different standard deviations of the gravity prior between 0.15 and 0.25*9.81 m/s$^2$ and the standard deviation of observed timing errors.

model is not activated for upside-down motion [32], a hypothesis which is also supported by our data.

Under this assumption, we can use the responses for the inverted gravity condition to estimate the errors introduced by motor variability. An inactivation of the gravity prior would mean that the gravity acting upon the object should be represented with the same precision as arbitrary gravities. We previously found Weber fractions of between 13% and beyond 30% for arbitrary gravities [40], which is in line with those found for linear accelerations [41]. We thus proceed with a value of 20%, which corresponds to a normalized standard deviation of 0.295 (see procedure above).

There are further constraints: First, the motor variability should be lower than the overall variabilities observed for the absolute error in each condition (the minimum is just over 0.08 s for the short occlusion condition with 1.3g and an initial vertical velocity of 4.5 m/s). Second, the motor variability should be equal across conditions and be independent of gravity, initial velocity and Occlusion category (see Fig 5B).

We put these values for **g**, $v_y$ and $d_y$ into Eq 7 to stimulate the temporal responses for each trial 1000 times. We minimize the Root Mean Square Errors (RMSE) between the standard deviations of the simulated timing error and the observed timing errors, separately for each combination of gravity, initial vertical velocity, Occlusion condition and participant. We collapsed the error across initial horizontal velocities because results for both values were virtually the same, mostly likely because the horizontal velocity barely influences overall flight duration in the presence of air drag, and not at all in the absence of air drag. After visualizing a relevant range of candidate values for the standard deviation of the remaining errors (see Fig 6), we use the optim() function implemented in R with a lower bound of 0.01 s and an upper bound of 0.06 s to find the best fit for the observed data. We found the best fit for a standard deviation of 0.058 s, with an RMSE of 0.04.

**The standard deviation of the gravity prior.** We then proceed to apply these values to simulate data sets based on the above assumptions, get the standard deviations for the timing error and compare them to standard deviations of the observed timing errors (Method 1). We restrict this comparison to the 0.7g/0.85g/1g/1.15/1.3g condition, as we expect the gravity model not to be activated for inverted gravitational motion. For a discussion of factors impacting the performance of the model for short occlusions, see [40]. We first simulate a range of sensible standard deviations (from 0, corresponding to an impossibly precise representation,

to 0.28, corresponding to a quite imprecise representation with limited impact on the final percept, in steps of 0.03) to determine the lower and upper bounds of the optimization interval (see Fig 6B). Fig 5D furthermore highlights how changes in the simulated variability of the represented gravity changes response variability.

We find the errors to be lowest around 0.21, and choose thus 0.16 as the lower bound and 0.26 m/s$^2$ as the upper bound. We then search for that standard deviation that minimizes the error between simulated and observed timing errors, using the optim() function implemented in R [31]. For each iteration, we simulate 1000 data sets and minimize the Root Mean Square Error (RMSE) between the standard deviations of simulated and observed timing errors across these 1000 data sets. The R code we used for these simulations can be found on GitHub (https://github.com/b-jorges/SD-of-Gravity-Prior), including extensive annotations. We found a normalized standard deviation of 0.208 for the gravity prior, which corresponds to a standard deviation of about 2.04 m/s$^2$ for a mean of 9.81 m/s$^2$, and a Weber fraction of 14.1%. The RMSE is 0.024. In Fig 7, we illustrate how the simulated standard deviations relate to the observed ones. The light red dots correspond to this method ("Simulated (Method1)"); as evident from the figure, the fits are better for the Long Occlusion condition, while the SDs are generally overestimated for the Short Occlusion condition.

If the gravity prior was discarded completely for upwards motion, we might observe even larger errors for -1g motion. We elaborate on this issue in the discussion. As there is thus some reason to believe that the gravity prior is not completely inactive in upwards motion, which may bias to above method to overestimate the standard deviation of the gravity prior, we furthermore conducted simulations where both the motor variability and the strong gravity prior are fitted to the data (Method 2). To this end, we use the optimize() function implemented in R which uses the Nelder and Mead method [42] to determine those values for the motor standard deviation and the standard deviation of the gravity prior that yield the smallest errors between simulated and observed variability. This is suitable because variability in the gravity prior and motor variability affect the final variability differentially (see Fig 5): a higher motor variability leads to uniformly higher standard deviations for the observed error, while a higher gravity variability affects longer trajectories (Long Occlusion, higher initial vertical velocity and lower gravities) more strongly than shorter ones. Based on above results, we chose 0.04 and 0.2 as starting parameters, but did not limit the parameter space. This method allots variability in slightly different proportions: the standard deviation for the motor error is 0.06 s and the standardized standard deviation of the gravity prior is 0.211 (which corresponds to a non-standardized standard deviation of 2.07 m/s$^2$ and a Weber fraction of 14.2%), with an RMSE of 0.024. These values are extremely close to the values found with Method 1. While it is worth noting that fitting both parameters to the data makes this method more susceptible to overfitting, this lends additional support to the tentative conclusion that the standard deviation of the gravity prior is just above 2 m/s$^2$ or a Weber Fraction of 14.2% The simulated standard deviations for these conditions are depicted in solid red in Fig 7 ("Simulated (Method 2)"): The fits are much better for the long occlusions, at the cost of a slight overestimation of the variability for the short occlusions.

## Discussion

Humans assume in many tasks and circumstances that objects in their environment are affected by earth gravity. It has thus been suggested that we maintain a representation of this value, which we then recruit to predict the behavior of objects in our environment. We recently interpreted this representation as a Strong Prior in a Bayesian framework [14]. A "Strong Prior" is a prior with a reliability so high that it overrules any sensory input

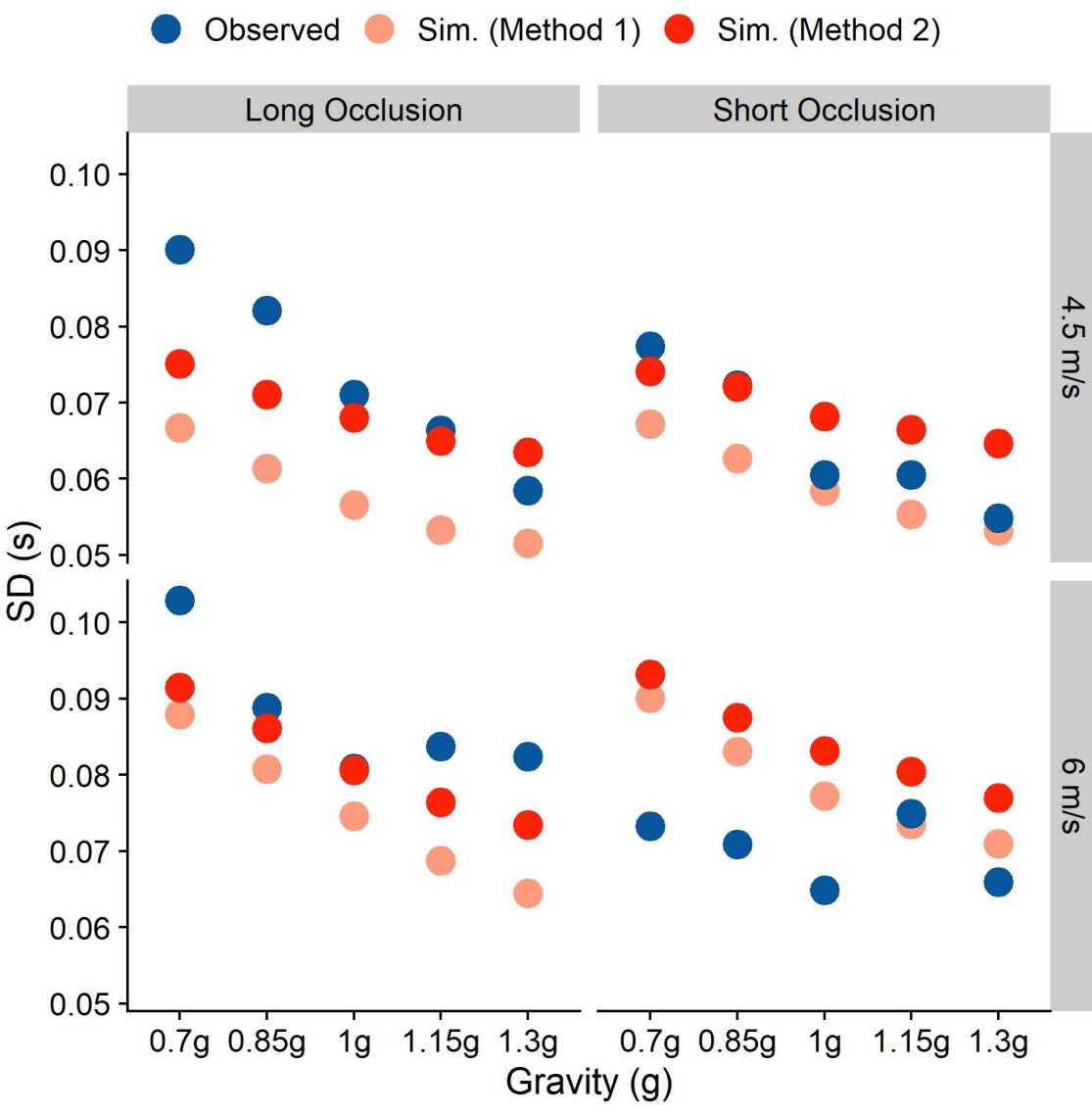

**Fig 7. Observed and simulated standard deviations separated by occlusion condition, initial vertical velocity and presented gravity.** Blue indicates the observed standard deviations across subjects, while the standard deviations simulated through the two-step process (Method 1) are coded light red and the standard deviations simulated through the two-parameter fit (Method 2) are coded solid red.

represented in the likelihood. Based on data from timing task (previously reported in [33]), we make an attempt at determining the standard deviation of a hypothetical Strong Earth Gravity Prior. Our general approach is to account for other sources of perceptuo-motor variability in the task based on thresholds reported in the literature, and attributing the remaining variability to the Gravity Prior. Based on this approach, we find a standard deviation of 2.13 m/s$^2$ (Method 1) or 2.07 m/s$^2$ (Method 2), for a prior with a mean of 9.81 m/s$^2$, which corresponds–mathematically–to a Weber fraction of 14.1% or 14.2%, respectively. This is considerably lower than Weber fractions generally observed for acceleration discrimination, but above Weber fractions for the discrimination of constant speeds [43].

Interestingly, when we simulated the timing errors with a fixed value of 9.81 m/s$^2$ (i.e., in a non-Bayesian framework where the value of earth gravity is not represented as a distribution,

but rather a value set at 1g; see [13] and also above), we found that our results fit the observed timing error quite nicely for each gravity value. That is, the observed gravity (corresponding to the Likelihood) had no discernable influence on the final percept (Posterior). However, in a Bayesian framework, this is only possible if the Likelihood is extremely shallow and the Prior is extremely precise. A Weber fraction of about 30% for the likelihood (which we assume for acceleration discrimination), and a Weber fraction of 14.1% or 14.2%. for the prior (as modelled) would not result in discarding the likelihood completely (see also Fig 1; even for a strong prior and a rather shallow likelihood, the likelihood attracts the posterior to some extent). Our results thus reveal a mismatch between the means observed in our experiment, the modelled standard deviation and a Bayesian explanation.

We see two possible ways to explain this mismatch. Firstly, our observed standard deviation for the gravity prior could be an upper bound. Our method relies on identifying all sources of variability and allotting variability in the response accordingly. Since we did not measure our participants' Weber fractions for velocity and distance discriminations individually, but rather used averages reported in the literature for somewhat different tasks, this may have distorted how much variability perceived distances and velocity at disappearance introduced in the response. Furthermore, when estimating the variability introduced in the motor response, we part from the premise that the internal model of gravity is not activated at all for -1g motion. However, we observe a bias to respond too late in this condition, suggesting that humans expect objects to accelerate less when moving upwards. This could be taken as evidence that the internal model of gravity is still activated to some extent. In this case, we would need to allot more variability to the motor error, which in turn would lead to a lower standard deviation for the gravity prior. However, this pattern in our data is also consistent with humans taking arbitrary accelerations into account insufficiently in perceptuo-motor tasks, which has been reported repeatedly for tasks where the gravity prior is highly unlikely to be recruited [41, 44–46]. The values of 14.1% or 14.2% obtained above may thus be an upper bound for the standard deviation of the Earth Gravity Prior.

A second possibility is that prior knowledge and online perceptual input are combined in a non-Bayesian fashion (and we should thus avoid the terminology "Prior", "Likelihood" and "Posterior"), where the mean of the final percept is set according to an acceleration of 9.81 m/s$^2$, while its standard deviation is determined by a (not necessarily Bayesian) combination of prior knowledge and online sensory information.

## Conclusion

In this paper, we build upon a simple model for coincidence timing of gravitational motion brought forward in [13]. By accounting for the Aubert-Fleischl phenomenon, we extend the domain of our model to also include shorter extrapolation intervals. Furthermore, we propose a procedure to determine the standard deviation of a potential gravity prior, and apply it to pre-existing data from a timing task. Standard deviations of 2.13 m/s$^2$ or 2.07 m/s$^2$ (depending on the method) explains the behavior observed in our task best. However, considering the literature we would expect an even lower standard deviation, as a Prior with a mean of 9.81 m/s$^2$ and standard deviations of 2.13 m/s$^2$ or 2.07 m/s$^2$ should not attract the Posterior as strongly as has been commonly observed. We thus believe that we are not able to fully disentangle different sources of noise in our data; the value we find for the standard deviation of the earth gravity prior is thus more likely an upper bound, and follow-up experiments may find lower values.

## Author Contributions

**Conceptualization:** Björn Jörges, Joan López-Moliner.

**Data curation:** Björn Jörges.

**Formal analysis:** Björn Jörges.

**Funding acquisition:** Joan López-Moliner.

**Investigation:** Björn Jörges, Joan López-Moliner.

**Methodology:** Björn Jörges.

**Software:** Björn Jörges.

**Supervision:** Joan López-Moliner.

**Validation:** Björn Jörges, Joan López-Moliner.

**Visualization:** Björn Jörges.

**Writing – original draft:** Björn Jörges.

**Writing – review & editing:** Björn Jörges, Joan López-Moliner.

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
