## [Decision Letter · Decision Letter 0]

14 Apr 2020

PONE-D-20-06855

Characterizing the Strong Earth Gravity Prior

PLOS ONE

Dear Mr. Joerges,

I hope you, your co-author, and the families are going well in this strange period.

Thank you for submitting your manuscript to PLOS ONE. After careful consideration, we feel that it has merit but does not fully meet PLOS ONE’s publication criteria as it currently stands. Therefore, we invite you to submit a revised version of the manuscript that addresses the points raised during the review process.

Having now received two reviews regarding your manuscript, I think I hardly could get reviews that are more different on your manuscript, and now stand in a very uncomfortable position to make a decision.

Where do we stand, just to act in full transparency: R1 sees no comment to make to your article, and would accept it as is, minus a very few details. On the contrary, R2 has many negative comments and recommend rejection. To make things even harder, it turns out this is not an independent manuscript, and you here re-analyze data that was published previously (which has call for a specific warning from the editorial team).

Regarding R2’s comments, I do not see any comment that would claim for a major issue, which would definitely call for an immediate rejection. I think all the comments are important, but you should have a chance to answer to them.

I am more concerned with the data reanalyze issue. I do not see a clear explanation why the current analysis was not proposed in the first article. Is there anything new, an article that got published, a new method etc… that led you to make the current analysis? At the moment, one may have the feeling that the current paper is just the second part of the first paper, and I think this is detrimental to its acceptance. Can it be specifically added why these analyses were not done initially, and why they are carried out now? Or something like the conclusion of the first paper seems at odd given you did not take into account the Aubert-Fleischl phenomenon, and you here attempt to answer this possible lack? I have been reviewing an article some time ago, Makin (2018), in which he re-analyze some of his own data (see specifically the supplementary part 2), and I think we miss an explanation like this currently.

Finally, having read myself your paper, a scientific comment as well:

We do not know how long the trajectory gets occluded in your experiment, depending on the occlusion ratio, which makes your results hard to compare to the literature. Doing myself TTC experiments, and using occlusion time within 0.5 - 3 s, I often get constant error between -1 s to +1 s, even so I mostly use constant velocities (the errors would be of a higher magnitude I think with acceleration). Hence, if you exclude trials with errors > .5 s and one participant because he has a mean error of .23 s, this is a major bias to me. I see no reason to exclude a participant because his performances are different from the others, this is clearly not an exclusion bias. I would exclude him because of the devices did not properly work, because he did not understood the task, forgot his glasses etc… Excluding trials in which the participants answered before the occlusion is perfectly fine to me for example. If I should use a performance criteria, I would choose one that cannot be questioned and it obviously not acceptable. In the experiments I am doing, I generally remove errors above 3 s. Here you are removing errors / participants that I would never dare considering as outliers. I think that by excluding these trials and participants, reducing the error toward 0 ms and the variability as you do, it is not surprising to confirm the participants perform well and confirm the existence of a 1g model. If existing, this model should deal with the complete variability of the data, not only on the selected trials.I would therefore strongly suggest to include these trials / participant, and see it this affect the outcome of the analysis.

Makin, A. D. J. (2018). The common rate control account of prediction motion. Psychonomic Bulletin & Review, 25, 1784-1797.

We would appreciate receiving your revised manuscript by May 29 2020 11:59PM. To enhance the reproducibility of your results, we recommend that if applicable you deposit your laboratory protocols in protocols.io, where a protocol can be assigned its own identifier (DOI) such that it can be cited independently in the future. For instructions see: http://journals.plos.org/plosone/s/submission-guidelines#loc-laboratory-protocols

We look forward to receiving your revised manuscript.

Kind regards,

Robin Baurès, Ph.D.

Academic Editor

PLOS ONE

Journal Requirements:

2. Please modify the title to ensure that it is meeting PLOS’ guidelines (https://journals.plos.org/plosone/s/submission-guidelines#loc-title). In particular, the title should be "specific, descriptive, concise, and comprehensible to readers outside the field" and in this case it is not informative and specific about your study's scope and methodology.

3. We noted in your submission details that a portion of your manuscript may have been presented or published elsewhere.

"We have published about this data before (Jörges & López-Moliner 2019). Our previous publication focussed on the eye-movement component of the project. We also presented the mean differences in timing errors and established a very simple model to account for these errors.

The present paper addresses a different research question than our previous publication: rather than comparing mean errors, we use the variability in responses to estimate how precisely humans represent the value of earth gravity.

We adapted the methods section for this manuscript to focus on the timing task. However, a strong overlap is unavoidable. Furthermore, Figure 2 from the present manuscripts was also used in the previous publication. We noted this in the manuscript."

4. Please ensure that you refer to Figure 2 in your text as, if accepted, production will need this reference to link the reader to the figure.

Reviewers' comments:

Reviewer's Responses to Questions

**Comments to the Author**

1. Is the manuscript technically sound, and do the data support the conclusions?

Reviewer #1: Yes

Reviewer #2: Partly

2. Has the statistical analysis been performed appropriately and rigorously? 

Reviewer #1: Yes

Reviewer #2: Yes

3. Have the authors made all data underlying the findings in their manuscript fully available?

Reviewer #1: Yes

Reviewer #2: Yes

4. Is the manuscript presented in an intelligible fashion and written in standard English?

Reviewer #1: Yes

Reviewer #2: Yes

5. Review Comments to the Author

Reviewer #1: The target manuscript aims to expand a Bayesian model, previously put forth by the authors, to account for timing errors of ball undergoing parabolic trajectories either in accordance or counter to earth's gravity. As an improvement from previous efforts, the authors explicitly include what could be expected from the Aubert-Fleischl effect. While doing so, the authors uncover a mismatch between the mean timings, the normal distribution used to model gravuty as a prior and a Bayesian explanation of the found variability. Overall, the manuscript is very interesting and well written. The assumptions for the modelizations are very well clarified and I found several of the steps particularly clever. I honestly do not think I can provide any feedback which would further improve this manuscript, apart from a few minor details:

1. Abstract, "while expands the range" - there seems to be an error in this sentence;

2. Page 3: While both the Bayes Theorem and the explanation of the Bayesian framework are clear by themselves, it would be better to link them more closely (e.g., by explicitly stating to which parameters in equation 1 does the prior [P(A)] and the Likelihood [P(B|A)/P(B)] refer to - I know this should be obvious for most readers, but it would help the readability of this section);

3. Page 8: "Figure 3: Temporal errors (...)" - this excerpt seems to be a figure caption. In fact, the first sentence reads the same as the caption for Figure 3. There is, however, an isolated sentence in this paragraph which do not seem to fit the text or the caption: "illustrated the distributions...". Please either remove these sentences or, if meant to convey what can be seen in the Figure, please reformulated it;

4. Page 11: "Figure 4 visualizes the mean errors" - I am not an english native speaker, but the verb "visualizes" sounds odd in this context (for a non-native speaker it sounds as if the Figure is activelly visualizing the mean errors); if this turns out to be a correct usage of the term, please ignore this comment;

Reviewer #2: The authors report an experiment testing the existence of a Strong Gravity Earth Prior (SGEP) in a coincidence timing task. The authors aim at determining the mean and the std of the SGEP. The submitted paper calls for two other articles already published by the authors, among which one reports one part of the collected data (eye-tracking) gained in a single experiment. The remaining data (temporal errors) are used in the present submitted paper. The first result of the experiment is that the Aubert-Fleischl phenomenon can account for the mean of SGEP. The second result of the experiment is that the variability of the SGEP can be retrieved from participants’ button-press temporal variability.

I agree that the work is for some aspects interesting. The experimental paradigm neither measurements are not very innovative but the scientific approach sounds good. Concerning the flow of the paper, the motivation to access the mean of SGEP can be easily understood for the non-specialist reader. Results can be easily understandable too. Indeed, there is a theoretical value of g and one can compare it to the participants' mean percept. The second goal of the experiment (“determine the standard deviation of the strong gravity prior” p.2) is for me much more obscure. Are there any examples of the measure of the standard deviation of Priors in the literature? What would be the expected value of it? This leads to a difficult interpretation of the authors' conclusions: “we are not able to fully disentangle different sources of noise in our data”.

Additionally, I do not like very much splitting a single experiment into several papers since it misleads the reader about the original experiment. Here, the reader can for instance not be aware of the experimental constraints (e.g., calibration process) linked to eye-tracking for participants that are however part of the experiment but too briefly reported. Moreover, this looks for me a rentability approach rather than a scientifically motivated approach. Temporal judgment and oculomotor behavior were linked, why spreading them? Finally, some paragraphs are strictly identical between papers without any further checks. For instance, can the authors justify that “The projectors introduced a delay of 0.049259 s (SD = 0.001894 s) “ (p.6)? I would be interested in an instrument able to measure events shorter than ms… I noticed several other points that at least request a cross-check. This gave to me a weird first impression of the work.

Experimentally, I’m very annoyed with several aspects of the experiment, despite some parts of the experimental protocol and data were already published otherwhere.

- My first concern is about the sample size. I can not understand how one of the authors can participate in such a psychological experiment and strongly suggest to remove its data. All participants must be naïve in most of the experiments in visual psychology. Additionally, I found the remaining sample of 8 participants (given that s9 was excluded from analysis, results sections p.9) too weak. The question looks to be already raised by a reviewer in the 2019’s paper since a “justification of sample size” paragraph appears in it. This, however, does not convince me. Finally, if the data used in the present paper were gained in a previous experiment (Jorges, Lopez-Moliner, 2019) as claimed p.4 “we use the data from our previous study (Jörges & López-Moliner, 2019)”, why are the gender of participants different between this paper (5 females) and the previous paper (3 females)?

- My second concern is about the number of experimental conditions (section procedure, p.6-7). It looks that participants performed 48 training trials + 3x320 trials + 64 trials for the “1 block of 1g/-1g motion" = 1168 trials. How long spent the experiment? Do you think that such an amount of trials did not let to a standardized perception? Do any participants report any fatigue or ennui?

- My third concern is about the repetition number. Since the paper focuses on the variability of prior, why not testing less trajectory but much more repetitions? You’re not dealing with 3D motion variability, just button-press. 8 repetitions of a temporal error are too low for supporting the authors' test about the variability of SGEP.

- My fourth concern addresses the data collection. First, can the authors guaranty that the mouse button can accurately measure temporal error? In such an experimental paradigm, high-temporal accuracy devices as E-Prime are usually required since the USB port and internal clock of the computer can delay the monitoring of USB mouse signal. Do the authors perform some test-bed ? Moreover, at no moment, the authors report the duration of the stimulus, which makes temporal data difficult to understand. Second, in the apparatus section (p. 6), why not offering any details about the eye-tracking device? post-processing methods? Type of dependent variable analyzed? This is an odd oversight because some interpretations resort to eye-tracking data (cf. simulations section, p10. “Our ad hoc explanation of this discrepancy was that subjects were often executing a saccade when the ball returned to initial height” “ and “Our subjects were specifically instructed to follow the target with their eyes, and the eye-tracking data we collected that they generally did pursue the target” ). The authors should at least mention their previous paper and report results.

I would finally like the author to carefully explain the ballistic of the ball trajectory and the related perceptual information available for perceiving g. Indeed, in a fully visible trajectory, g can be retrieved from different sources. In their experiments, authors occlude some parts of the trajectory. Naïve readers must understand what information remains available from all information usually available. This must also be connected to real-world illustrations, that must convince the reader that humans usually have to succeed in performing such tasks (e.g. in sports for instance) and that the experimental paradigm can mirror human perceptual processes

Minor comments :

- Participant section (p.4), why “remaining participants”?

- Apparatus section (p. 6). Does the virtual scene be enslaved to the participants' viewpoint? Perhaps the variability of judgments is related to a change in perception of the altitude?

- Stimuli section (p.4). Does the ball spin during its trajectory?

- Stimuli Section (p.5). please details why the -6.15m depth from the observer was chosen and the perceptual consequences of such a parameter (FOV, role of stereoscopy, usual perceptual processes operating at this range of distance, etc.)

- I found the paragraph “On a theoretical level […] to recover its physical velocity from retinal motion” (p.3) very crude, without any theoretical references to any framework while it is connected to a specific field of visual psychology. Optic Flow carries a lot of visual information that alone can be used to disambiguate environment perception for ecological psychologists. Also, for cyberneticians, “precise estimate of the observed world” is achieved through internal models. Please add references and pragmatic examples that support the authors' claims.

- Result section (p.8). The reference to Figure 3 is weirdly inserted in connection to the following explanation in the text. Please correct.

- Result section (p.8). The “following test model” looks to be identical to the first equation (p.7). Why going back regarding the previous equation?

- Result section (p.9). I would prefer expressing the temporal errors as a function of the duration of the trajectory to figure out their magnitude.

- Result section. Please provide all data corresponding to the statistical tests (especially effect size).

- The sentence “However, in the case of gravity it seems that the expectation of Earth Gravity overrules all sensory information that humans collect on the law of motion of an observed object. “(p3-4) looks to be a claim without any experimental evidence.

- The Figure 1 is so much underexploited in the text that unfamiliar readers with Bayesian Theory might be loose in interpreting it.

- The reference to “figure 1” in the Section “stimuli”, p.5 looks to be incorrect in the paragraph that refers to the virtual scene (cf. figure 2)

- A missing figure reference had to be corrected in p. 17

- “we expect the gravity model not to be activated for inverted gravitational motion” (p. 16) looks to contradicts with “there is some reason to believe that the gravity prior is not completely inactive in upwards motion” (P. 17)

6. PLOS authors have the option to publish the peer review history of their article (what does this mean?). If published, this will include your full peer review and any attached files.

Reviewer #1: Yes: Nuno De Sá Teixeira

Reviewer #2: No

---

## [Author Response · Author response to Decision Letter 0]

11 May 2020

Response to the editor

I am more concerned with the data reanalyze issue. I do not see a clear explanation why the current analysis was not proposed in the first article. Is there anything new, an article that got published, a new method etc… that led you to make the current analysis? At the moment, one may have the feeling that the current paper is just the second part of the first paper, and I think this is detrimental to its acceptance. Can it be specifically added why these analyses were not done initially, and why they are carried out now? Or something like the conclusion of the first paper seems at odd given you did not take into account the Aubert-Fleischl phenomenon, and you here attempt to answer this possible lack? I have been reviewing an article some time ago, Makin (2018), in which he re-analyze some of his own data (see specifically the supplementary part 2), and I think we miss an explanation like this currently.

The first paper addressed the question whether the internal representation of gravity was used to guide eye-movements and motion extrapolation (see also our preregistration at https://osf.io/8vg95/). We also introduced a very simple model, which, however, only predicted responses for trials with a long occlusion. It broke down for shorter occlusions and we suggested a possible explanation for this lacking fit for shorter occlusions (an interaction with saccades). Importantly, we never touched upon variability in timing responses, mostly because the main thrust of this project were eye-movements. 

This second project uses a different approach (modelling and simulations) to answer a different research question: what are the parameters of this strong earth gravity prior? From the literature and our previous analysis, it was quite clear that the mean of this prior had to be around 9.81 m/s². However, it hadn’t been attempted to characterize the standard deviation of this prior. What we see as main contribution of this paper is thus both the method by which we obtain the standard deviation and the standard deviation itself.

It is important to add that we only included the methods and a repetition of a part of the results for the convenience of the reader. We realized that this could give the impression that we were presenting (new) experimental results. We reduced the methods to the bare minimum and removed the biggest part of the results from the resubmission, and added a note that readers that were interested in experimental design and accuracy results could consult our earlier publication. The contributions we aim to make with this manuscript are, as stated above, the refinement of the model and the simulations that enable us to extract the standard deviation of the gravity prior.

We do not know how long the trajectory gets occluded in your experiment, depending on the occlusion ratio, which makes your results hard to compare to the literature. Doing myself TTC experiments, and using occlusion time within 0.5 - 3 s, I often get constant error between -1 s to +1 s, even so I mostly use constant velocities (the errors would be of a higher magnitude I think with acceleration). Hence, if you exclude trials with errors > .5 s and one participant because he has a mean error of .23 s, this is a major bias to me. I see no reason to exclude a participant because his performances are different from the others, this is clearly not an exclusion bias. I would exclude him because of the devices did not properly work, because he did not understood the task, forgot his glasses etc… Excluding trials in which the participants answered before the occlusion is perfectly fine to me for example. If I should use a performance criteria, I would choose one that cannot be questioned and it obviously not acceptable. In the experiments I am doing, I generally remove errors above 3 s. Here you are removing errors / participants that I would never dare considering as outliers. I think that by excluding these trials and participants, reducing the error toward 0 ms and the variability as you do, it is not surprising to confirm the participants perform well and confirm the existence of a 1g model. If existing, this model should deal with the complete variability of the data, not only on the selected trials. I would therefore strongly suggest to include these trials / participant, and see it this affect the outcome of the analysis.

Applying the criterion of an absolute error of < 0.5 after removing subject 9 results only in a loss of 269 trials, that is 2.2% of the remaining trials. That is, this criterion was relatively liberal. Please note that our occlusion times are between 0.2 and 0.8 s, that is, quite short (we also added this information to the manuscript).

Regarding subject 9, we have the strong worry that their performance was influenced heavily by other biases (due to VR presentation); as visible from the (adjusted) Figure 4 in the manuscript, their performance differs quite extremely from the other subjects (they are the only participant whose error ratio consistently lies above 2, in comparison to the other participants who barely ever exceed 1.25). For the resubmission, we have, nonetheless, included s09 (but excluded the author, s10, as per Reviewer 2’s request) and adjusted the exclusion criterion to 2 seconds, which represents about 250 % of the longest occlusion duration. This led to removal of 216 trials or 1.7% of all trials that remained after exclusion of the author’s data. 

 

Response to the Reviewers

Reviewer #1: 

The target manuscript aims to expand a Bayesian model, previously put forth by the authors, to account for timing errors of ball undergoing parabolic trajectories either in accordance or counter to earth's gravity. As an improvement from previous efforts, the authors explicitly include what could be expected from the Aubert-Fleischl effect. While doing so, the authors uncover a mismatch between the mean timings, the normal distribution used to model gravuty as a prior and a Bayesian explanation of the found variability. Overall, the manuscript is very interesting and well written. The assumptions for the modelizations are very well clarified and I found several of the steps particularly clever. I honestly do not think I can provide any feedback which would further improve this manuscript, apart from a few minor details:

1. Abstract, "while expands the range" - there seems to be an error in this sentence;

Addressed, thank you.

2. Page 3: While both the Bayes Theorem and the explanation of the Bayesian framework are clear by themselves, it would be better to link them more closely (e.g., by explicitly stating to which parameters in equation 1 does the prior [P(A)] and the Likelihood [P(B|A)/P(B)] refer to - I know this should be obvious for most readers, but it would help the readability of this section);

Thanks for raising this point. We expanded the whole section.

3. Page 8: "Figure 3: Temporal errors (...)" - this excerpt seems to be a figure caption. In fact, the first sentence reads the same as the caption for Figure 3. There is, however, an isolated sentence in this paragraph which do not seem to fit the text or the caption: "illustrated the distributions...". Please either remove these sentences or, if meant to convey what can be seen in the Figure, please reformulated it;

Thank you, addressed.

4. Page 11: "Figure 4 visualizes the mean errors" - I am not an english native speaker, but the verb "visualizes" sounds odd in this context (for a non-native speaker it sounds as if the Figure is activelly visualizing the mean errors); if this turns out to be a correct usage of the term, please ignore this comment;

Addressed.

Reviewer #2: 

The authors report an experiment testing the existence of a Strong Gravity Earth Prior (SGEP) in a coincidence timing task. The authors aim at determining the mean and the std of the SGEP. The submitted paper calls for two other articles already published by the authors, among which one reports one part of the collected data (eye-tracking) gained in a single experiment. 

We have published one paper based on this dataset (in Scientific Reports), not two. The first paper contains analyses of both behavioral tasks (ocular pursuit and timing responses). After publication of that paper, we became interested in characterizing the gravity prior not only in terms of its mean, but also in terms of its standard deviation, and realized that our published data could be used for this purpose. While the simulations proposed in this manuscript are of course related to the previous analyses, we believe that we go beyond (re-)reporting the behavioral data, both in terms of its research question (“can we fully characterize the gravity prior?” rather than “Do humans rely on gravity to guide eye-movements and interceptive timing?”) and its modelling/simulation-based (rather than behavioral) approach.

The remaining data (temporal errors) are used in the present submitted paper. The first result of the experiment is that the Aubert-Fleischl phenomenon can account for the mean of SGEP. The second result of the experiment is that the variability of the SGEP can be retrieved from participants’ button-press temporal variability. I agree that the work is for some aspects interesting. The experimental paradigm neither measurements are not very innovative but the scientific approach sounds good. Concerning the flow of the paper, the motivation to access the mean of SGEP can be easily understood for the non-specialist reader. Results can be easily understandable too. Indeed, there is a theoretical value of g and one can compare it to the participants' mean percept. The second goal of the experiment (“determine the standard deviation of the strong gravity prior” p.2) is for me much more obscure. Are there any examples of the measure of the standard deviation of Priors in the literature? What would be the expected value of it? This leads to a difficult interpretation of the authors' conclusions: “we are not able to fully disentangle different sources of noise in our data”.

There are studies on optimal integration of prior information with online evidence. These studies typically manipulate the standard deviation of the prior in virtue of the stimulus distribution they represent. We, in turn, are attempting to measure the standard deviation of a prior that is formed by our interactions with the natural environment. “Natural” priors have been studied before, e.g., a slow motion prior, a light-from-above prior or a bigger-is-heavier prior (Adams, Graf, & Ernst, 2004; Peters, Ma, & Shams, 2016; Stocker & Simoncelli, 2006; Thornton & Lee, 2000). However, to our knowledge, we are the first ones to fully characterize such prior in terms of its mean and standard deviation. We expanded on our explanation of the Bayesian framework to make clearer what motivated our hypothesis about the standard deviation.

We expect the standard deviation of this prior to be very low. This is because only a very narrow prior can attract the mean of the posterior as strongly as reported in the literature, and as we found in our experiment (see analysis in our Scientific Reports paper). If we were pressed to put a number on our prior expectation of its standard deviation, we would choose 10% of its mean (i.e., 1 m/s²). I.e., it would be represented more precisely than linear velocities, which are represented with a standard deviation of 10-15% of the respective means. We added a sentence specifying this expectation in the introduction.

Additionally, I do not like very much splitting a single experiment into several papers since it misleads the reader about the original experiment. Here, the reader can for instance not be aware of the experimental constraints (e.g., calibration process) linked to eye-tracking for participants that are however part of the experiment but too briefly reported. Moreover, this looks for me a rentability approach rather than a scientifically motivated approach. Temporal judgment and oculomotor behavior were linked, why spreading them? 

We understand the reviwer's concern and want to make clear that none of the authors of this manuscript like “slicing” or producing more than one paper from a single project. We are aware that this impression might arise. The motivation for the present simulations arose after the behavioral results were already published. The focus of this manuscript goes beyond the scope of the previous paper, and we believe that the use of published datasets is acceptable if the paper adds sufficiently novel results. In our opinion, both the modelling-based procedure we use to recover the standard deviation of the strong earth gravity prior and its tentative value satisfy this condition of novelty. Please note that we did not split up ocular behavior and manual judgments in two papers. We report the results for both tasks in the previous paper. The present manuscript, in turn, focusses on computational aspects behind these behavioral results.

Please note that we re-described the methods for the convenience of the reader. We now realize that this might lead readers to believe that these were new experimental results. This was not our intention and we apologize. We reduced the methods to the bare minimum necessary to understand the task and added a reference to the previous paper.

Finally, some paragraphs are strictly identical between papers without any further checks. For instance, can the authors justify that “The projectors introduced a delay of 0.049259 s (SD = 0.001894 s) “ (p.6)? I would be interested in an instrument able to measure events shorter than ms… I noticed several other points that at least request a cross-check. This gave to me a weird first impression of the work.

The delay refers to the relative difference between the projector's output and audio output measured by an analog oscilloscope HAMEG HM 1505 which works with a resolution of 150 MHz. The estimation of the relative delay is based on 100 samples. The value reported here corresponds to the mean across all of these trials. We adjusted the manuscript to reflect that this is a mean value.

Experimentally, I’m very annoyed with several aspects of the experiment, despite some parts of the experimental protocol and data were already published otherwhere.

- My first concern is about the sample size. I can not understand how one of the authors can participate in such a psychological experiment and strongly suggest to remove its data. All participants must be naïve in most of the experiments in visual psychology. Additionally, I found the remaining sample of 8 participants (given that s9 was excluded from analysis, results sections p.9) too weak. The question looks to be already raised by a reviewer in the 2019’s paper since a “justification of sample size” paragraph appears in it. This, however, does not convince me. 

While certainly not the most recommended approach, being one’s own participant is, in our understanding, not an uncommon situation in our field. It is generally assumed that there is not enough voluntary control over performance, and in fact, the author’s performance is very similar to the other participants’ performance (if somewhat less variable, possibly due to their experience with the experiment). However, we are very sympathetic to this concern and repeat the simulations without the author’s data. As per the editor’s request, we include s09, which brings us back to an n of 9. The results are very similar.

The power analysis (reported in the justification of sample size section) was conducted before data collection (please see also our preregistration under https://osf.io/8vg95/). While we agree that a higher n is always desirable, we found under quite conservative assumptions that a sample size of 10 was sufficient to detect effects. Note that the power analysis was conducted with the eye-tracking task in mind, not for the time-to-contact estimation. However, considering the effect sizes observed in the timing task, 9 participants (with 1344 trials each) would be enough to achieve a power of nearly 1. This is, of course, assuming that the inter-subject variability we find in our dataset is representative of the variability in the overall population. 

Finally, if the data used in the present paper were gained in a previous experiment (Jorges, Lopez-Moliner, 2019) as claimed p.4 “we use the data from our previous study (Jörges & López-Moliner, 2019)”, why are the gender of participants different between this paper (5 females) and the previous paper (3 females)?

That is was a mistake in the current manuscript and we apologize for this oversight. The numbers from the published paper are correct. We corrected this mistake.

- My second concern is about the number of experimental conditions (section procedure, p.6-7). It looks that participants performed 48 training trials + 3x320 trials + 64 trials for the “1 block of 1g/-1g motion" = 1168 trials. How long spent the experiment? Do you think that such an amount of trials did not let to a standardized perception? Do any participants report any fatigue or ennui?

The experiment was about one hour of testing overall. Calibrating the eye-tracker before the session and between the blocks added on average another 30 minutes to the experiment. The participants did report being bored by the task at times, but dividing the experiment up in four blocks meant that they could take breaks every 15 minutes. 

We address the issues of a regression to the mean/central tendency, which may be a consequence of standardized perception, in the Scientific Reports paper; in brief, we present three pieces of evidence: (1) errors from trials with different gravities but the same time-to-contact should be biased strongly towards a common mean error. This is not the case; there is a bias, but it is very small. (2) The overall pattern of responses fit our gravity model much better than a central tendency model. (3) variability in our data is heavily correlated with flight time; if participants used the same response criterion independently of presented trajectory, variability should be equal across conditions.

- My third concern is about the repetition number. Since the paper focuses on the variability of prior, why not testing less trajectory but much more repetitions? You’re not dealing with 3D motion variability, just button-press. 8 repetitions of a temporal error are too low for supporting the authors' test about the variability of SGEP.

We present each combination of stimulus variables 24 times overall: eight times in each of the three 0.7g-1.3g blocks. In the -1g/1g condition, we only had two different gravity values and thus managed to accommodate all 24 repetitions in one block. Furthermore, differences in contact timing were negligible between both initial horizontal velocities (very likely because it barely impacts the flight duration). For this reason, we collapsed trials and obtained thus the standard deviations across 48 trials for each condition combination (gravity x initial vertical velocity x Occlusion category). We realize that this was ambiguous in the manuscript and adjusted it accordingly.

- My fourth concern addresses the data collection. First, can the authors guaranty that the mouse button can accurately measure temporal error? In such an experimental paradigm, high-temporal accuracy devices as E-Prime are usually required since the USB port and internal clock of the computer can delay the monitoring of USB mouse signal. Do the authors perform some test-bed ? 

Comercial stimulus presentation software like E-Prime or presentation are not suitable for 3D complex stimulus presentations. Usually, programs needs to be writen in C or python (like in our case). We use the openGl engine in python (pyglet) devoted to gaming, which aims to reach maximum precision both for stimulus frames and input recording. We access the mouse time stamps directly iohub python libraries (which merges with psychopy) which circumvents the main system events loop and uses the clock_gettime(CLOCK_MONOTONIC) in unix-like systems (like os x, the one we use). The precision is sub-miliseconds. Iohub can be used with or without psychopy realtime access to input devices. Importantly, it runs its own thread devoted to continuously sampling the input device state independently of the video (stimulus) thread.

Moreover, at no moment, the authors report the duration of the stimulus, which makes temporal data difficult to understand. 

This was indeed an oversight, which we corrected. Thank you!

Second, in the apparatus section (p. 6), why not offering any details about the eye-tracking device? post-processing methods? Type of dependent variable analyzed? This is an odd oversight because some interpretations resort to eye-tracking data (cf. simulations section, p10. “Our ad hoc explanation of this discrepancy was that subjects were often executing a saccade when the ball returned to initial height” “ and “Our subjects were specifically instructed to follow the target with their eyes, and the eye-tracking data we collected that they generally did pursue the target”). The authors should at least mention their previous paper and report results.

We did not want to repeat results from our previous paper excessively (for the reasons the reviewer outlined above); we added references to the previous paper to the passages in question.

I would finally like the author to carefully explain the ballistic of the ball trajectory and the related perceptual information available for perceiving g. Indeed, in a fully visible trajectory, g can be retrieved from different sources. In their experiments, authors occlude some parts of the trajectory. Naïve readers must understand what information remains available from all information usually available. This must also be connected to real-world illustrations, that must convince the reader that humans usually have to succeed in performing such tasks (e.g. in sports for instance) and that the experimental paradigm can mirror human perceptual processes

Minor comments :

- Participant section (p.4), why “remaining participants”?

This was an error, we tested 10 participants overall. We apologize for the oversight.

- Apparatus section (p. 6). Does the virtual scene be enslaved to the participants' viewpoint? Perhaps the variability of judgments is related to a change in perception of the altitude?

We did not adjust the stimulus to the height of the participants. After excluding the author, the tallest participant was about 1.80 and the shortest about 1.65. At a 6 m difference to the stimulus, the differences in angle between the eyes of the tallest and the shortest participant are at most 2°. While this may have some influence on between-participant variability, any effect should be cancelled out due to the within-participant design of the experiment.

- Stimuli section (p.4). Does the ball spin during its trajectory?

The ball does not spin. Since we reduced the methods part greatly, we did not add this information to the manuscript.

- Stimuli Section (p.5). please details why the -6.15m depth from the observer was chosen and the perceptual consequences of such a parameter (FOV, role of stereoscopy, usual perceptual processes operating at this range of distance, etc.)

We mostly cut the description of the methods to minimize overlap with what has been published before, which would include this information.

To answer the reviewer’s questions nonetheless: when looking at the center of the screen, the field of view was 50° horizontally and 62° vertically, and stimuli were presented in a range of about 40° horizontally and 20° vertically. 

For the task at hand, binocular cues are mostly relevant to estimating the distance of the object. While they are quite noisy at a distance of 6.15 m, the high accuracy in responses suggests that our participants managed to recover the distance correctly, probably using prior knowledge about the target (López-Moliner & Keil, 2012), the internal model of gravity (Lacquaniti et al., 2015) and the visual environment. In our task, binocular cues are nearly constant throughout the trajectories, which is why it is very likely that participants rely mostly on monocular cues.

- I found the paragraph “On a theoretical level […] to recover its physical velocity from retinal motion” (p.3) very crude, without any theoretical references to any framework while it is connected to a specific field of visual psychology. Optic Flow carries a lot of visual information that alone can be used to disambiguate environment perception for ecological psychologists. Also, for cyberneticians, “precise estimate of the observed world” is achieved through internal models. Please add references and pragmatic examples that support the authors' claims.

We added references for the concept of “Encoding/Decoding”. We furthermore made our theoretical commitments more explicit and expanded on our (previously very rudimentary) example.

- Result section (p.8). The reference to Figure 3 is weirdly inserted in connection to the following explanation in the text. Please correct.

Thank you, we corrected this mistake.

- Result section (p.8). The “following test model” looks to be identical to the first equation (p.7). Why going back regarding the previous equation?

These equations referred to different types of analysis (accuracy vs. precision). However, since we cut the accuracy analysis to minimize overlap, the first one is no longer needed.

- Result section (p.9). I would prefer expressing the temporal errors as a function of the duration of the trajectory to figure out their magnitude.

We transformed the error to ratios ((error + occluded time)/occluded time).

- Result section. Please provide all data corresponding to the statistical tests (especially effect size).

To minimize overlap with our previous paper, we cut the analysis regarding means. Furthermore, we transformed the results of the Bayesian analyses from the log space into normal space, which should make them much more interpretable; especially together with Table 1, which lists the standard deviations for each condition including variability that the Bayesian Mixed Model assigned to each individual. (This is why the values in the table are higher than the values reported for the model.)

- The sentence “However, in the case of gravity it seems that the expectation of Earth Gravity overrules all sensory information that humans collect on the law of motion of an observed object. “(p3-4) looks to be a claim without any experimental evidence.

We added references to substantiate this claim.

- The Figure 1 is so much underexploited in the text that unfamiliar readers with Bayesian Theory might be loose in interpreting it.

We expanded on the explanation of the Bayesian framework and relied more on Figure 1.

- The reference to “figure 1” in the Section “stimuli”, p.5 looks to be incorrect in the paragraph that refers to the virtual scene (cf. figure 2)

Corrected, thank you.

- A missing figure reference had to be corrected in p. 17

We are not sure which figure reference the reviewer is referring to exactly, but we carefully checked all references to make sure that all are in order.

- “we expect the gravity model not to be activated for inverted gravitational motion” (p. 16) looks to contradicts with “there is some reason to believe that the gravity prior is not completely inactive in upwards motion” (P. 17)

This issue required a longer explanation, and we were under the impression that it was not the right place to elaborate. Instead, we dedicated a paragraph of the discussion to the issue. We added a reference to that section in the place in question.

Bibliography:

Adams, W. J., Graf, E. W., & Ernst, M. O. (2004). Experience can change the “light-from-above” prior. Nature Neuroscience, 7(10), 1057–1058. https://doi.org/10.1038/nn1312

Lacquaniti, F., Bosco, G., Gravano, S., Indovina, I., La Scaleia, B., Maffei, V., & Zago, M. (2015). Gravity in the Brain as a Reference for Space and Time Perception. Multisensory Research, 28(5–6), 397–426. https://doi.org/10.1163/22134808-00002471

López-Moliner, J., & Keil, M. (2012). People Favour Imperfect Catching by Assuming a Stable World. Current Science, (4), 1435–1439. https://doi.org/10.1371/Citation

Peters, M. A. K., Ma, W. J., & Shams, L. (2016). The Size-Weight Illusion is not anti-Bayesian after all: a unifying Bayesian account. PeerJ, 4, e2124. https://doi.org/10.7717/peerj.2124

Stocker, A. A., & Simoncelli, E. P. (2006). Noise characteristics and prior expectations in human visual speed perception. Nature Neuroscience, 9(4), 578–585. https://doi.org/10.1038/nn1669

Thornton, A., & Lee, P. (2000). Publication bias in meta-analysis: Its causes and consequences. Journal of Clinical Epidemiology, 53(2), 207–216. https://doi.org/10.1016/S0895-4356(99)00161-4

---

## [Decision Letter · Decision Letter 1]

2 Jul 2020

PONE-D-20-06855R1

Determining Mean and Standard Deviation of the Strong Gravity Prior through Simulations

PLOS ONE

Dear Dr. Joerges,

Thank you for submitting your manuscript to PLOS ONE. After careful consideration, we feel that it has merit but does not fully meet PLOS ONE’s publication criteria as it currently stands. Therefore, we invite you to submit a revised version of the manuscript that addresses the points raised during the review process.

As you will see in the reviews, I could possibly not receive more opposed reviews from the two reviewers. To help me getting a decision, I have contacted and been help by another academic editor that I wish to thank here, who has read and commented the reviews. We both agree that you have made substantial modifications that were asked by the second reviewer, in particular re-included S09 and excluded S10, which appears unnoticed by the second reviewer. It however appears that in referring to the previous article to describe the experimental design, you might have over-reacted. We feel that with the current article alone, it would be hard to a reader to understand the task. We should think that all readers might not have access to the previous publication, and hence the description of the task should stand on its own in the present manuscript. 

Finally, the second reviewer still disagrees with the use of the mouse and the keyboard to collect temporal response. It seems to both me and the additional editor that with cautions on the software side, which you did, these two devices remain appropriate to collect these answers. However, I would recommend adding a few lines to present the reviewer's concern, with the citation, so that any reader can shape his own mind on the debate.

We look forward to receiving your revised manuscript.

Kind regards,

Robin Baurès, Ph.D.

Academic Editor

PLOS ONE

Reviewers' comments:

Reviewer's Responses to Questions

**Comments to the Author**

1. If the authors have adequately addressed your comments raised in a previous round of review and you feel that this manuscript is now acceptable for publication, you may indicate that here to bypass the “Comments to the Author” section, enter your conflict of interest statement in the “Confidential to Editor” section, and submit your "Accept" recommendation.

Reviewer #1: All comments have been addressed

Reviewer #2: (No Response)

2. Is the manuscript technically sound, and do the data support the conclusions?

Reviewer #1: Yes

Reviewer #2: No

3. Has the statistical analysis been performed appropriately and rigorously? 

Reviewer #1: Yes

Reviewer #2: Yes

4. Have the authors made all data underlying the findings in their manuscript fully available?

Reviewer #1: Yes

Reviewer #2: Yes

5. Is the manuscript presented in an intelligible fashion and written in standard English?

Reviewer #1: Yes

Reviewer #2: No

6. Review Comments to the Author

Reviewer #1: The authors successfully addressed all the points raised in my previous review. I do believe that the current version of the manuscript is fit for publication.

Reviewer #2: I'm disappointed in this manuscript. The introduction has been reworked, the methods removed, the first part of the results reanalyzed with a more suitable dependent variable, and the discussion has been kept almost intact. Given the abundance of errors in the previous version, I would have expected a cautious verification by the authors, but errors persist in the references of the figures (cf. lines 83, 91, 143, 145), in the typing errors (cf. line 67). I found it very disturbing to hide all the information about the methods (e.g., participants, task), especially because they hide important experimental questions raised in the previous expertise. This hides the nature of the participants. In response to the authors, I urge them to distance themselves from colleagues who commonly integrate themselves as participants in their psychological experiences. It is absolutely necessary to avoid the authors being part of the population in psychological experiments in order to avoid introducing certain voluntary and involuntary biases in the behaviour of the experimenters. In the revised manuscript, it is not clear that the experimenters are part of the data analysed. In addition, the deletion of the method section hides the instruments used to monitor participant response. I continue to argue that the mouse and keyboard are not good tools for recording temporal responses despite the argument provided. Please read Plant & Turner, Behavior Research Methods, 2009. This makes the publication methodologically unacceptable. Finally, this deletion has led the authors to merge information regarding the exclusion of trials in the results section. I don't understand why the inclusion/exclusion of about 1% of the trials seems to be a necessity for the authors. What is behind all these efforts?

7. PLOS authors have the option to publish the peer review history of their article (what does this mean?). If published, this will include your full peer review and any attached files.

Reviewer #1: **Yes: **Nuno Alexandre De Sá Teixeira

Reviewer #2: No

---

## [Author Response · Author response to Decision Letter 1]

10 Jul 2020

Editor:

"As you will see in the reviews, I could possibly not receive more opposed reviews from the two reviewers. To help me getting a decision, I have contacted and been help by another academic editor that I wish to thank here, who has read and commented the reviews. We both agree that you have made substantial modifications that were asked by the second reviewer, in particular re-included S09 and excluded S10, which appears unnoticed by the second reviewer. It however appears that in referring to the previous article to describe the experimental design, you might have over-reacted. We feel that with the current article alone, it would be hard to a reader to understand the task. We should think that all readers might not have access to the previous publication, and hence the description of the task should stand on its own in the present manuscript."

-- We have extended the Methods section to reflect all relevant information.

"Finally, the second reviewer still disagrees with the use of the mouse and the keyboard to collect temporal response. It seems to both me and the additional editor that with cautions on the software side, which you did, these two devices remain appropriate to collect these answers. However, I would recommend adding a few lines to present the reviewer's concern, with the citation, so that any reader can shape his own mind on the debate."

-- We have included Reviewer 2’s concern in the manuscript and have explained our strategy to mitigate delays and variability added by the input device.

 

Reviewer 2:

"Given the abundance of errors in the previous version, I would have expected a cautious verification by the authors, but errors persist in the references of the figures (cf. lines 83, 91, 143, 145), in the typing errors (cf. line 67)."

-- We apologize. Some of these errors were introduced into the manuscript in the process of formatting the document for resubmission. We addressed these errors in this version.

"I found it very disturbing to hide all the information about the methods (e.g., participants, task), especially because they hide important experimental questions raised in the previous expertise. This hides the nature of the participants."

-- We had reduced the Methods section to minimize overlap with the published paper. As per the reviewer’s and the editor’s request, we have re-included this information in this version of the manuscript.

"In response to the authors, I urge them to distance themselves from colleagues who commonly integrate themselves as participants in their psychological experiences. It is absolutely necessary to avoid the authors being part of the population in psychological experiments in order to avoid introducing certain voluntary and involuntary biases in the behaviour of the experimenters. In the revised manuscript, it is not clear that the experimenters are part of the data analysed."

-- We thank the reviewer for their advice. As stated in our previous response, we are highly in agreement with this concern and have removed the author from our analyses. The overall number of included participants remains the same (n = 9) because we re-included one participant we had previously excluded based on the editor’s feedback.

"In addition, the deletion of the method section hides the instruments used to monitor participant response. I continue to argue that the mouse and keyboard are not good tools for recording temporal responses despite the argument provided. Please read Plant & Turner, Behavior Research Methods, 2009. This makes the publication methodologically unacceptable."

-- As per the editor’s suggestion, we are making possible constraints as well as our mitigation efforts much clearer in the resubmitted version of the manuscript.

"Finally, this deletion has led the authors to merge information regarding the exclusion of trials in the results section. I don't understand why the inclusion/exclusion of about 1% of the trials seems to be a necessity for the authors. What is behind all these efforts?"

-- These trials include, for example, trials where the program froze, participants did not pay attention or other situations that led to extremely large recorded errors. These excluded trials do not reflect participant performance in any meaningful way.

---

## [Editor Report · Decision Letter 2]

14 Jul 2020

Determining Mean and Standard Deviation of the Strong Gravity Prior through Simulations

PONE-D-20-06855R2

Dear Dr. Joerges,

We’re pleased to inform you that your manuscript has been judged scientifically suitable for publication and will be formally accepted for publication once it meets all outstanding technical requirements.

Kind regards,

Robin Baurès, Ph.D.

Academic Editor

PLOS ONE
---

## [Editor Report · Acceptance letter]

23 Jul 2020

PONE-D-20-06855R2 

Determining Mean and Standard Deviation of the Strong Gravity Prior through Simulations 

Dear Dr. Jörges:

I'm pleased to inform you that your manuscript has been deemed suitable for publication in PLOS ONE. Congratulations! Your manuscript is now with our production department. 

Kind regards, 

on behalf of

Dr. Robin Baurès 

Academic Editor

PLOS ONE